# Unveiling the Magic of Code Reasoning through Reflective Hypothesis Decomposition and Amendment

**Yuze Zhao**[1], **Tianyun Ji**[1,*], **Wenjun Feng**[1,*], **Zhenya Huang**[1,2,†], **Qi Liu**[1,2],
**Zhiding Liu**[1], **Yixiao Ma**[1], **Kai Zhang**[1], **Enhong Chen**[1]
[1]State Key Laboratory of Cognitive Intelligence,
University of Science and Technology of China
[2]Institute of Artificial Intelligence, Hefei Comprehensive National Science Center
`yuzezhao@mail.ustc.edu.cn huangzhy@ustc.edu.cn`

## Abstract

The reasoning abilities are one of the most enigmatic and captivating aspects of large language models (LLMs). Numerous studies are dedicated to exploring and expanding the boundaries of this reasoning capability. However, tasks that embody both reasoning and recall characteristics are often overlooked. In this paper, we introduce such a novel task, **code reasoning**, to provide a new perspective for the reasoning abilities of LLMs. We summarize three meta-benchmarks based on established forms of logical reasoning, and instantiate these into eight specific benchmark tasks. Our testing on these benchmarks reveals that LLMs continue to struggle with identifying satisfactory reasoning pathways. Additionally, we present a new pathway exploration pipeline inspired by human intricate problem-solving methods. This **R**eflective **H**ypothesis **D**ecomposition and **A**mendment (**RHDA**) pipeline consists of the following iterative steps: (1) Proposing potential hypotheses based on observations and decomposing them; (2) Utilizing tools to validate hypotheses and reflection outcomes; (3) Revising hypothesis in light of observations. Our approach effectively mitigates logical chain collapses arising from forgetting or hallucination issues in multi-step reasoning, resulting in performance gains of up to $3\times$. Finally, we expand this pipeline by applying it to simulate complex household tasks in real-world scenarios, specifically in VirtualHome, enhancing the handling of failure cases. We release our code and all of results at `https://github.com/TnTWoW/code_reasoning`.

## 1 Introduction

Large Language Models (LLMs), which are trained on billions of tokens, have demonstrated impressive reasoning abilities in complex tasks (Brown et al., 2020; Wei et al., 2022; Kojima et al., 2022; OpenAI, 2023). However, it is evident that as potential fuzzy retrieval systems or parameterized knowledge compression systems (Xie et al., 2021), LLMs perform better on System 1 tasks than on System 2 tasks (Kahneman, 2011; Bengio, 2019; Yao et al., 2023a; Weston & Sukhbaatar, 2023; Liu et al., 2023a). Specifically, LLMs excel in intuitive memory retrieval tasks, but continue to face significant challenges with tasks requiring rational reasoning (Kambhampati, 2024).

From the perspective of human cognitive psychology, **reasoning can be viewed as a process of memory retrieval**, in which people retrieve relevant information from memory and use it to make inferences (Kyllonen & Christal, 1990; Süß et al., 2002; Hayes et al., 2014; Feeney & Thompson, 2014; Hardman & Cowan, 2015). For example, Haidt (2001) proposed that when individuals engage in moral reasoning, they typically draw on their prior knowledge from social and cultural contexts. Similarly, studies involving animal lesions and human neuroimaging have confirmed that the hippocampus, which is primarily associated with memory, also plays a crucial role in reasoning

---

[*]Equal contribution
[†]Corresponding author

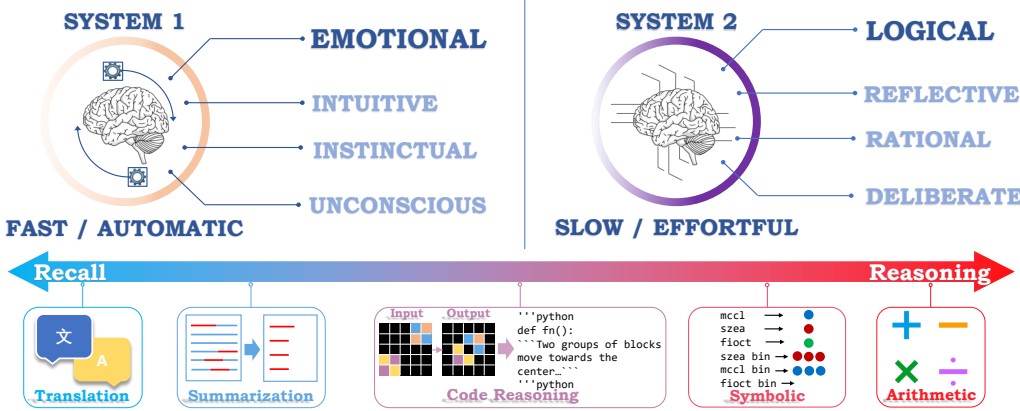

Figure 1: Code reasoning is a category of tasks that incorporates logical reasoning into code, aiming to solve programming problems through logical reasoning. These tasks require a balance between background knowledge and thinking span, placing greater emphasis on the collaborative functioning of both System 1 and System 2 thinking.

abilities (Zeithamova et al., 2012). Therefore, memory and reasoning are interdependent, with considerable overlap between the two, rendering the distinction between them somewhat arbitrary (Heit et al., 2012; Liu et al., 2023b).

We believe that, similar to humans (Strachan et al., 2024; Liu et al., 2024b; Lin et al., 2024), LLMs do not exhibit a clear boundary between memory and reasoning (Schaeffer et al., 2024; Razeghi et al., 2022). However, tasks that lie at this intersection are often overlooked in research. Here, we propose a novel task to explore the capability boundaries of LLMs: **Code Reasoning**. Code reasoning encompasses a category of tasks that demonstrate logical reasoning through code and address problems in a systematic manner. As illustrated in Figure 1, we position some tasks along an axis that reflects 1) the degree of reliance on prior knowledge (Recall) and 2) the extent to which prior knowledge is applied to the current context (Reasoning). We position the code reasoning task between memory and reasoning. On one hand, the highly structured nature of code requires the model to learn syntax from pre-training data, enabling it to recall relevant information during solving a problem. On the other hand, generating code solutions necessitates the model's understanding of the problem and context, involving reasoning to produce appropriate solutions.

In this paper, we introduce code reasoning, a task that formalizes reasoning steps into a programming language and offloads the computation process to the compiler. To explore different aspects of code reasoning, we summarize three meta-benchmarks based on existing forms of logical reasoning: inductive code reasoning, deductive code reasoning, and abductive code reasoning.

Inductive code reasoning involves deriving broad generalizations from a series of observations, demonstrating the ability to infer rules from examples and generate programs to meet input-output mapping. Deductive code reasoning starts from premises and derives valid conclusions, focusing on the model's capacity to understand a program's intermediate states and reasoning step by step. Abductive code reasoning seeks the simplest and most likely explanation based on a set of observations, highlighting the model's ability to abstractly understand a function's purpose.

We concretize these three meta-benchmarks into eight specific benchmarks. Based on these eight benchmarks, we evaluate the performance of existing models in code reasoning. Due to data sparsity, we find that current state-of-the-art LLMs still struggle to achieve satisfactory results in solving such problems. To enhance the reasoning process, we implement a **R**eflective **H**ypothesis **D**ecomposition and **A**mendment (**RHDA**) pipeline. This pipeline is iterative, encompassing hypothesis decomposition, execution verification, and amendment submission. Specifically, we first guide the LLM to formulate initial hypotheses based on complex observations and decompose these into sub-hypotheses. These sub-hypotheses are then compiled into executable functions through a translator, enabling direct application to the observations, followed by validation using external tools. Subsequently, based on the execution results and observations, the LLM submits amendments to reflect on and refine the issues within the sub-hypotheses.

Our experimental results indicate that RHDA methods effectively mitigate reasoning failures caused by data sparsity. With the same or even lower overhead, this method achieved performance improvements of up to three times compared to baseline methods. Finally, we extend this pipeline to complex, simulated real-world household tasks VirtualHome (Puig et al., 2018; 2020), guiding the LLM to complete a series of intricate operations.

## 2 META-BENCHMARK

We describe the general process of code reasoning as the transformation from Input $\mathcal{I}$ and Program $\mathcal{P}$ to Output $\mathcal{O}$, represented as $\mathcal{I} \xrightarrow{\mathcal{P}} \mathcal{O}$. Inductive code reasoning is concretized as the Programming by Example (PBE) task. In this task, a neural program synthesis model $\mathcal{M}$ searches the execution space to find a program that best satisfies all given input-output specifications. We donate this meta-benchmark as $\mathcal{M}(\mathcal{I}, \mathcal{O}) \to \widetilde{\mathcal{P}}$. Deductive code reasoning is exemplified in tasks that simulate the program execution process. In this task, a neural simulation compiler model $\mathcal{M}$ tracks the program's execution and records intermediate states, gradually deriving the final valid output. We denote this meta-benchmark as $\mathcal{M}(\mathcal{I}, \mathcal{P}) \to \widetilde{\mathcal{O}}$. Abductive code reasoning is concretized as input prediction tasks. This task requires the neural understanding model $\mathcal{M}$ to form an abstract-level understanding of function's behavior and perform abductive inference based on the given program and output. We represent this meta-benchmark as $\mathcal{M}(\mathcal{O}, \mathcal{P}) \to \widetilde{\mathcal{I}}$. The details of the benchmarks are provided in the Appendix C.

### 2.1 INDUCTIVE CODE REASONING

Inductive code reasoning can be represented as $\mathcal{M}(\mathcal{I}, \mathcal{O}) \to \widetilde{\mathcal{P}}$ and is concretized as a PBE task (Qiu et al., 2024; Shi et al., 2024). PBE is a program synthesis task designed to help end-users, particularly non-programmers, create scripts to automate repetitive tasks (Gulwani, 2016). Based on input-output specifications, PBE systems can synthesize a program in either a general-purpose language (GPL) or a domain-specific language (DSL). Inductive code reasoning encompasses four challenging PBE tasks, two of which are GPL tasks: List Function (Rule, 2020) and MiniARC (Kim et al., 2022), while the other two are DSL tasks: RobustFill (Devlin et al., 2017) and DeepCoder (Balog et al., 2016). GPL tasks are relatively complex, allowing the model to solve problems in a more flexible manner. In contrast, DSL tasks require the model to quickly learn the syntax of DSL through few-shot learning and address relatively simpler problems.

**List Function.** The List Function task was originally designed to investigate how humans learn the concept of computable functions that map lists to lists. Given input and output specifications in the form of lists, the model generates GPL rules that conform to these specifications. For example, with an input specification of `[2, 4, 8, 10]` and an output specification of `[3, 5, 9, 11]`, we expect the resulting rule to be `lambda x : x + 1`[1].

**MiniARC.** MiniARC is a compressed 5x5 version of the Abstraction and Reasoning Corpus (Chollet, 2019; Moskvichev et al., 2023), designed to assess imaginative and reasoning abilities. MiniARC balances the length of the input-output pairs with the difficulty of the problems. The specifications are 5x5 2D grids, where the numbers represent blocks of specific colors. The model must find valid problem-solving paths (such as color swapping, row flipping) to achieve the transformation from input to output.

**RobustFill.** RobustFill is a string manipulation task where the model is expected to perform a combination of atomic operations, such as extracting a substring from position $k_1$ to $k_2$ using $SubString(k_1, k_1)$, to achieve generalization. As an example, a program `ToCase(Lower, SubStr(1,3))` converts full month names (January, April) to their abbreviations (jan, apr).

**DeepCoder.** The DeepCoder task involves using DSL to perform operations on integer lists. In DeepCoder, each line represents a subroutine that performs atomic operations on previous variables and assigns the results to new variables. The result of the final line is the program's out-

---

[1] For conciseness while maintaining generality, we will use lambda expressions to represent a program.

put. For example, program `a ← [int] | b ← FILTER(<0) a | c ← MAP(*4) b | d ← SORT c | e ← REVERSE b` (where "|" denotes subroutine separator.) transforms the input `[-17, -3, 4, 11, 0, -5, -9, 13, 6, 6, -8, 11]` into the output `[-12, -20, -32, -36, -68]`. We provide detailed RobustFill and Deepcoder DSLs in Appendix A.

## 2.2 DEDUCTIVE CODE REASONING

Deductive code reasoning refers to the process of deriving a sound inference $\mathcal{O}$ by reasoning from the given premise $\mathcal{I}$, assuming the validity of the argument $\mathcal{P}$. Deductive code reasoning can be instantiated as an output prediction task (Gu et al., 2024). Based on the given premise, the output prediction requires the LLM to simulate a compiler (Kim et al., 2024b), executing step by step until it arrives at a valid conclusion. For example, given a program `P = lambda text, value: ''.join(list(text) + [value])` and inputs `text = 'bcksrut'`, `b = 'q'`, the output prediction from LLM should be `'bcksrutq'`.

## 2.3 ABDUCTIVE CODE REASONING

Starting from existing facts $\mathcal{P}$ and $\mathcal{O}$, deriving the most reasonable and optimal explanation $\mathcal{I}$ is referred to as abductive code reasoning. This meta-benchmark can be framed as an input prediction task. Given the provided facts, the input prediction requires the LLM to backtrack through the program's execution process to recover the potential inputs. In cases where multiple possible inputs exist, the model should apply Occam's Razor and return the simplest input. For example, given a program `P = lambda nums: nums + [nums[i % 2] for i in range(len(nums))]` and outputs `[-1, 0, 0, 1, 1, -1, 0, -1, 0, -1]`, the input prediction from LLM should be `[-1, 0, 0, 1, 1]`.

Deductive code and abductive code reasoning can be regarded as opposite processes; therefore, we selected two identical and representative datasets, CRUXEval (Gu et al., 2024) and LiveCodeBench (Jain et al., 2024), as benchmarks to validate these two capabilities.

**CRUXEval.** CRUXEval is a benchmark designed to evaluate code understanding and execution. Many models that achieve high scores on HumanEval (Chen et al., 2021) do not show the same level of improvement on the CRUXEval benchmark. This benchmark includes 800 functions along with their corresponding inputs and outputs.

**LiveCodeBench.** LiveCodeBench is a dynamically updated benchmark sourced from competition platforms. Each problem is timestamped, and we selected data from October 2023 (later than GPT-4o training) to March 2024 (the most recent), ensuring there is no data leakage and thereby guaranteeing the model's generalization performance.

## 3 CODE REASONING WITH HYPOTHESIS DECOMPOSITION AND AMENDMENT

We aim to generate a reliable reasoning process for problem-solving by establishing a problem-solving pathway $f : \mathcal{X} \rightarrow \mathcal{Y}$. For a given task $\tau$ and the seen specifications/observations $\mathcal{X}_\tau^s$, the pathway $f$, should lead to a seen valid solution $\mathcal{Y}_\tau^s$ through a chain of reasoning. We expect this pathway $f$ to have sufficient generalization capabilities to handle unseen specifications/observations $\mathcal{X}_\tau^u$. To this aim, we employ a process involving hypothesis decomposition, execution verification, and amendment submission to iteratively explore and refine the reasoning pathway. We first establish an initial hypothesis $h^0 \in \Sigma^*$ based on observations $x_\tau^s \in \mathcal{X}_\tau^s$, where $\Sigma^*$ is the closure form of LLM's vocabulary. This initial hypothesis $h^0$ serves as a preliminary solution pathway to the problem. Given the complexity of many problems, we decompose the hypothesis $h^0$ into simpler sub-hypotheses $h^0 \iff \{h_{s_0}^0, h_{s_1}^0, h_{s_2}^0, ...\}$. A translator function $g : \Sigma^* \rightarrow \Sigma_\mathcal{E}^*$, which maps the hypothesis space $\Sigma^*$ into an executable function space $\Sigma_\mathcal{E}^*$, is then used to 'compiled' the sub-hypotheses $h^0$ into an executable function $e^0$. This executable function is directly applicable to the observations $x_\tau^s$, allowing for the derivation of conclusions $\widetilde{y}_\tau^s$, that is:

$$\widetilde{y}_\tau^s = g(h^0)(x_\tau^s). \tag{1}$$

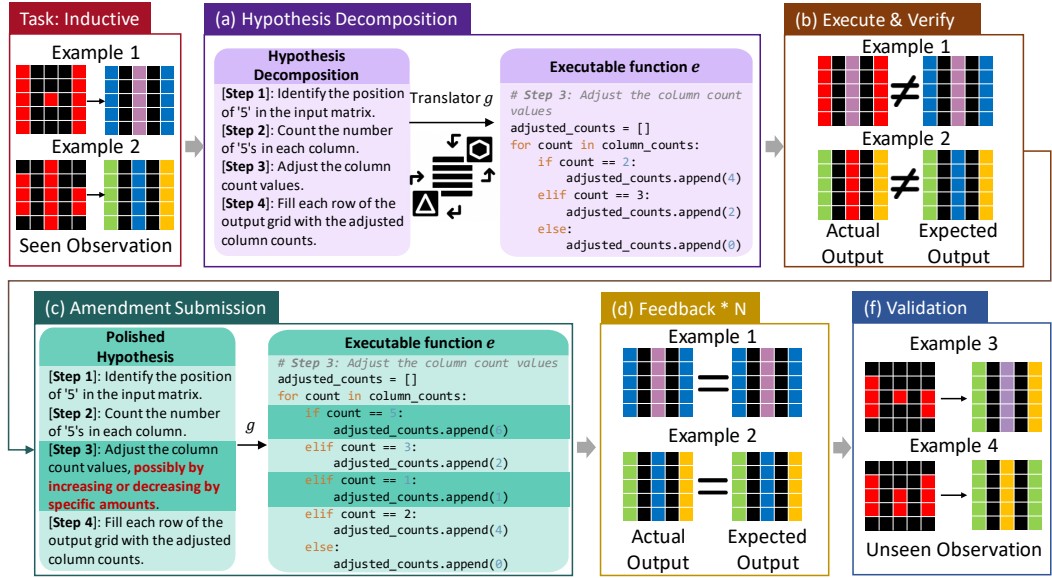

Figure 2: An overview of pipeline to solve code reasoning task. We decompose the hypothesis and generate executable functions step by step. After comparing the results with the seen observations and receiving feedback, we propose amendments, reflect on potential errors at each step, and generate revised hypotheses. This process is repeated until a valid problem-solving pathway is discovered. For concise expression, we show partial code snippets.

Feedback $\mathcal{F}(y_\tau^s, \tilde{y}_\tau^s)$ is used to evaluate the conclusions drawn from the current hypothesis, guiding the LLM to reflect on its sub-hypotheses. Through this iterative process of reflection, the model generates a new hypothesis $h^1$ for the next iteration. Finally, the problem-solving pathway $f$ is applied to unseen observations $\mathcal{X}_\tau^u$, and the model's generalization performance is assessed by measuring its accuracy:

$$acc_\tau = \frac{1}{|\mathcal{X}_\tau^u|} \sum_{x_\tau^u \in \mathcal{X}_\tau^u} \mathbf{1}\left[f(x_\tau^u) = y_\tau^u\right]. \tag{2}$$

The preceding section presents a unified framework for the hypothesis decomposition and amendment method. However, the implementation specifics differ across various tasks. In the following sections, we will introduce these task-specific variations in detail.

**Hypothesis Decomposition.** We recognize that complex logical reasoning problems are difficult to encapsulate in a single reasonable hypothesis, which can adversely affect the performance of LLMs. Therefore, we require the LLM to decompose its hypotheses. Specifically, given an observation $x_\tau^s$, the LLM gradually presents corresponding hypotheses step by step. For inductive code reasoning, $h_0$ represents the step-by-step hypothesis of the input-to-output transformation rules. For deductive and abductive code reasoning, $h_0$ refers to the step-by-step hypothesis regarding the functionality of the program.

**Execution Verification.** After obtaining the hypothesis, we need to apply it to the observations. However, hypotheses are often not directly usable, so we need to convert the decomposed hypothesis into an executable function $e$ through a translator $g$. For inductive code reasoning, the executable function is a program; for deductive and abductive code reasoning, the executable function is the predicted output and input, respectively. These three types of task are then sent to a compiler to obtain the actual execution results, and the feedback generated by the compiler is provided to the LLM to help it further refine and adjust the sub-hypotheses.

**Amendment Submission.** During the amendment submission stage, there are no significant differences in handling the three tasks. The LLM receives validation feedback from the tools and generates amendments based on this feedback, reflecting on possible issues in the previous hypothe-

Table 1: RHDA method on inductive code reasoning task. $T$ refers to the maximum number of iterations. $N$ refers to the number of candidates.

| Method | Accuracy | | | | Task Accuracy | | | |
|---|---|---|---|---|---|---|---|---|
| | List Func | MiniARC | RobustFill | Deepcoder | List Func | MiniARC | RobustFill | Deepcoder |
| IO | **64.85** | **28.21** | **61.74** | 23.78 | 38.00 | 13.08 | 21.74 | 10.42 |
| PoT | 44.90 | 10.90 | 37.39 | 30.90 | 33.60 | 8.46 | 21.74 | 19.79 |
| CoC | 42.45 | 10.90 | 31.30 | 26.39 | 34.40 | 4.62 | 13.04 | 13.54 |
| SC (N=3) | 52.95 | 12.31 | 46.09 | 37.85 | 41.20 | 9.23 | 26.09 | 26.04 |
| SR (T=2) | 51.10 | 10.26 | 41.74 | 36.81 | 41.60 | 8.46 | 21.74 | 25.00 |
| w/o Sub-Hyp | 42.45 | 7.95 | 40.87 | 18.05 | 33.20 | 4.62 | 21.74 | 9.37 |
| w/o Amend | 47.10 | 8.46 | 35.65 | 30.21 | 36.40 | 6.92 | 17.39 | 19.79 |
| T=2, N=1 | 51.05 | 12.56 | 43.48 | 38.89 | 41.20 | 10.77 | 30.43 | 23.96 |
| T=3, N=1 | 53.20 | 14.10 | 47.83 | 38.19 | 44.00 | 11.54 | 30.43 | 26.04 |
| T=2, N=3 | 58.35 | 19.74 | 54.78 | **43.06** | **48.80** | **13.85** | **34.78** | **29.17** |

ses. The reflection process involves revising each sub-hypothesis individually, forming an updated hypothesis $h_1 \iff \{h_{s_0}^1, h_{s_1}^1, h_{s_2}^1, ...\}$. This process ensures that each sub-hypothesis is adjusted to better align with the observations and validation results, gradually improving the reasoning pathway's coherence and accuracy.

## 4 EXPERIMENTS

**Experimental Setup.** We utilize the latest and most advanced model, gpt-4o-2024-08-06, as the backbone LLM for all our experiments. We report the results using Llama-3.1-70B-Instruct, Qwen-max (qwen-max-2024-09-19) (Bai et al., 2023), Claude 3.5 (claude-3-5-sonnet-20240620) in Appendix B. Following the methodology of Qiu et al. (2024), we set the temperature to 0.7. We report results using several methods: input-output (IO) prompting, standard prompting, Chain of Thought (CoT) (Wei et al., 2023), Program of Thought (PoT) (Chen et al., 2023), Chain of Code (CoC) (Li et al., 2024), Self-Consistency (SC) (Wang et al., 2023c) and Self-Refine (SR) (Madaan et al., 2024), all implemented with 2-shot learning.[2] For our proposed process, we employ 0-shot prompts, allowing the LLM to explore problem-solving pathways in a more flexible manner. We provide detailed prompt templates in Appendix H.

### 4.1 INDUCTIVE CODE REASONING

For inductive code reasoning, we establish four baseline methods. The Input-Output (IO) prompting requires the LLM to predict outputs based on all seen observations and an unseen input. The Program of Thought (PoT) method generates and executes programs to derive outputs. The CoC method prompts the LLM to utilize pseudocode for reasoning in output prediction. The SC method builds upon PoT by sampling multiple programs and selecting the one that demonstrates optimal performance on seen observations. Furthermore, since each example may contain multiple unseen observations, we adopt the approach from (Qiu et al., 2024) to define task accuracy externally. An example is deemed passed only when all unseen observations within it pass; thus, the proportion of passed examples reflects the task accuracy. The experimental results are presented in Table 1.

The results demonstrate that the RHDA method achieves optimal performance across four benchmarks, with task accuracy exceeding that of the second-best methods by 18.45%, 5.89%, 33.31%, and 12.02%, respectively. However, we observe that RHDA appears to underperform compared to IO prompting. This is because the IO prompt does not generate a hypothesis that satisfies all observations but instead predicts the output for a single input. A successful prediction for a single instance does not generate a hypothesis that satisfies all observations, resulting in a high prediction accuracy but a relatively low task accuracy.

**Ablation Study.** We introduce two variants to separately validate the effectiveness of hypothesis decomposition and amendment submission. The first variant does not require the LLM to decompose

---

[2]Not all methods are suitable for these three meta-benchmarks, thus we selected the most appropriate methods for each benchmark.

hypotheses, referred to as w/o Sub-Hyp. The second variant, termed w/o Amend, indicates that the model no longer modifies its hypotheses through reflection. The experimental results presented in Table 1 show that the performance of these two variants declined by 25.39% to 67.88% and 19.28% to 57.14%, respectively. This finding suggests that the introduction of sub-hypotheses is a critical step, as it simplifies complex problems, reducing the workload for the subsequent translator $g$ while also enabling individual amendments to each sub-hypothesis. Nonetheless, the reflection process is equally important. Our results align with previous research (Zhao et al., 2024; Olausson et al., 2024; Peng et al., 2023) indicating that rational reflection can significantly enhance performance.

## 4.2 DEDUCTIVE CODE REASONING

For deductive code reasoning, we select standard prompting, CoT, SC, SR and CoC as benchmark methods. The experimental results are presented in Table 2. These results indicate that the CoT and CoC methods significantly enhanced the accuracy of reasoning outcomes by guiding the model to think step-by-step about function capabilities. Our proposed method advances this further, achieving optimal performance with a single round of amendments, resulting in an improvement of up to 104.37% compared with baseline method. A horizontal comparison of the two datasets revealed that, due to the absence of LiveCodeBench data in internet corpora, the performance with standard prompts showed a marked advantage, with the SC method amplifying this gap. Notably, the combination of CoT, CoC, and hypothesis decomposition and amendment enabled the LLM to exhibit a substantial degree of reasoning and generalization ability, nearly solving all presented problems.

Table 2: RHDA method on deductive code reasoning task. $T$ refers to the maximum number of iterations. $N$ refers to the number of candidates.

|  | CRUXEval | LiveCodeBench |
| --- | --- | --- |
| Standard | 68.75 | 41.18 |
| CoT | 89.12 | 83.14 |
| SC (N=3) | 71.12 | 36.27 |
| SR (T=2) | 80.38 | 63.73 |
| CoC | 85.62 | 81.37 |
| w/o Amend | 86.62 | 71.29 |
| T=2, N=1 | **90.62** | **84.16** |

## 4.3 ABDUCTIVE CODE REASONING

For abductive code reasoning, we employ the same baseline methods as those used for deductive reasoning. The experimental results are presented in Figure 3. Compared to deductive reasoning, abductive reasoning involves a reverse thinking process, which presents significant challenges. The LLM cannot derive the program's intermediate states through deduction and must first establish an abstract-level understanding of the function's behavior before proceeding with abduction. On the CRUXEval dataset, the performance decline for abductive reasoning ranged from 8.20% to 25.52%. However, the hypothesis decomposition and amendment approach demonstrate robustness, as the change in reasoning modes resulted in only minimal performance degradation (8.20%) while still outperforming baseline methods by 10.02% to 31.89% on the CRUX-Eval dataset and 7.35% to 40.39% on the LiveCodeBench dataset. A horizontal comparison of the two datasets revealed a trend similar to that observed in deductive reasoning, with an overall performance decline on the LiveCodeBench dataset, suggesting a complex relationship between reasoning and recall.

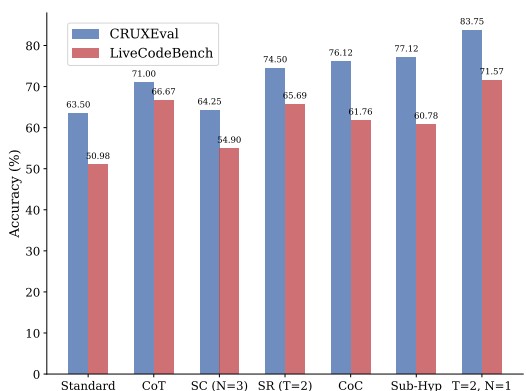

Figure 3: RHDA method on abductive code reasoning task. $T$ refers to the maximum number of iterations. $N$ refers to the number of candidates.

## 4.4 QUALITATIVE ANALYZE

We select some cases to conduct an in-depth exploration of the quality of RHDA.

Table 3: We compare the results obtained using the sub-hypothesis decomposition method with those obtained without it. The results without hypothesis decomposition are presented at the top of the table, while those with hypothesis decomposition are shown below. Benchmark: MiniARC-ID26.

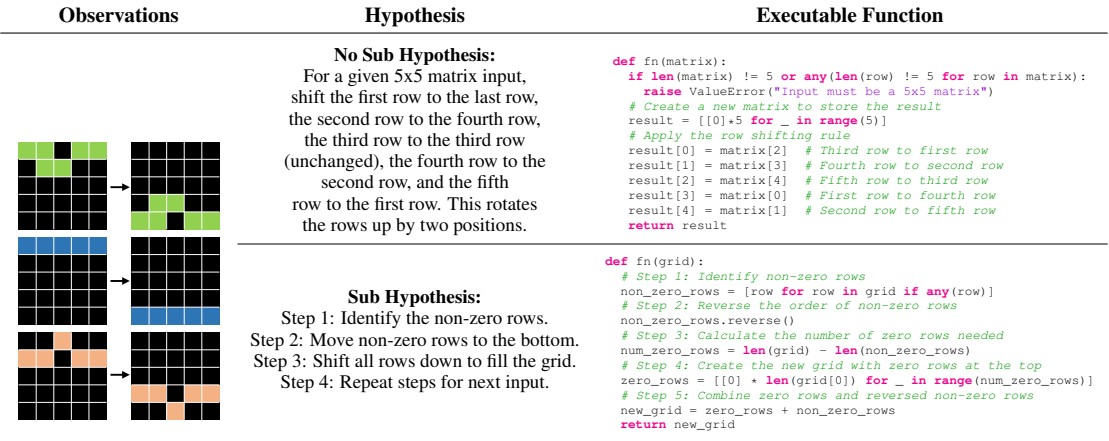

| Observations | Hypothesis | Executable Function |
|---|---|---|
| | **No Sub Hypothesis:**
For a given 5x5 matrix input, shift the first row to the last row, the second row to the fourth row, the third row to the third row (unchanged), the fourth row to the second row, and the fifth row to the first row. This rotates the rows up by two positions. | ```def fn(matrix):\n  if len(matrix) != 5 or any(len(row) != 5 for row in matrix):\n    raise ValueError("Input must be a 5x5 matrix")\n  # Create a new matrix to store the result\n  result = [[0]*5 for _ in range(5)]\n  # Apply the row shifting rule\n  result[0] = matrix[2]  # Third row to first row\n  result[1] = matrix[3]  # Fourth row to second row\n  result[2] = matrix[4]  # Fifth row to third row\n  result[3] = matrix[0]  # First row to fourth row\n  result[4] = matrix[1]  # Second row to fifth row\n  return result``` |
| | **Sub Hypothesis:**
Step 1: Identify the non-zero rows.
Step 2: Move non-zero rows to the bottom.
Step 3: Shift all rows down to fill the grid.
Step 4: Repeat steps for next input. | ```def fn(grid):\n  # Step 1: Identify non-zero rows\n  non_zero_rows = [row for row in grid if any(row)]\n  # Step 2: Reverse the order of non-zero rows\n  non_zero_rows.reverse()\n  # Step 3: Calculate the number of zero rows needed\n  num_zero_rows = len(grid) - len(non_zero_rows)\n  # Step 4: Create the new grid with zero rows at the top\n  zero_rows = [[0] * len(grid[0]) for _ in range(num_zero_rows)]\n  # Step 5: Combine zero rows and reversed non-zero rows\n  new_grid = zero_rows + non_zero_rows\n  return new_grid``` |

**Hypothesis Decomposition Reduces Task Complexity.** In Table 3, we present a qualitative analysis of the MiniARC benchmark, comparing the effects of using hypothesis decomposition versus not using it. As illustrated in the examples above, without hypothesis decomposition, the descriptions require a substantial amount of language, which ultimately leads to severe hallucinations in the program and results in failure. In contrast, the examples below, which utilize hypothesis decomposition, present clear and executable objectives, enabling the translator to easily write executable functions and achieve successful solutions as instructed.

Table 4: We present the additional improvements resulting from the use of amendments. The results shown above are those obtained after the initial hypotheses, while the results displayed below reflect the outcomes following the submission of amendments and subsequent reflections. Benchmark: List Function-ID29.

| Observations | Rounds | Executable Function |
|---|---|---|
| [0, 8, 3, 9] → [0, 8]
[6, 1] → []
[4, 8, 7] → [4]
... | **Round 1:**
Step 1: Remove elements from the end of the input list until a '9' is encountered.
Step 2: If '9' is present, remove all elements after the last occurrence of '9'.
Step 3: If '9' is not present, remove the last two elements of the list.
Step 4: If the list has fewer than two elements, return an empty list. | ```def fn(lst):\n  # Check if the list has fewer than two elements\n  if len(lst)<2:\n    return []\n  # Check if '9' is present in the list\n  if 9 in lst:\n    # Find the last occurrence of '9'\n    last_nine_index=len(lst)-1-lst[::-1].index(9)\n    # Return the list up to and including the last '9'\n    return lst[:last_nine_index+1]\n  else:\n    # If '9' is not present,\n    # remove the last two elements\n    return lst[:-2]``` |
| | **Round 2:**
Step 1: Remove the last two elements from the input list.
Step 2: If the resulting list has fewer than two elements, return an empty list. | ```def fn(input_list):\n  # Remove the last two elements from the input list\n  modified_list = input_list[:-2]\n  # Check if the list has fewer than two elements\n  if len(modified_list) < 2:\n    return []\n  return modified_list``` |

**Amendments Guide LLM Towards Correct Pathway.** We present a qualitative analysis of the use of amendments in the List Function benchmark in Table 4. The upper section displays the initialization of the hypothesis, where the LLM generates a potential guess based on the observations and translates it into an executable program. After offloading the execution to the tool (e.g., Python executor) and receiving feedback, amendments are proposed to modify the initial hypothesis. Following this reflection, the LLM re-optimizes the rules, ultimately yielding the correct execution results. More qualitative analyse examples please refer to Appendix E.1.

**Failure Analyse.** We also conduct an in-depth analysis of the reasons behind process failures in RHDA, detailed in Appendix E.2. Our findings reveal that the primary limitation arises from the restricted intrinsic reasoning capabilities of LLMs, which continue to face challenges in understanding and addressing complex problems. These limitations are primarily reflected in two aspects:

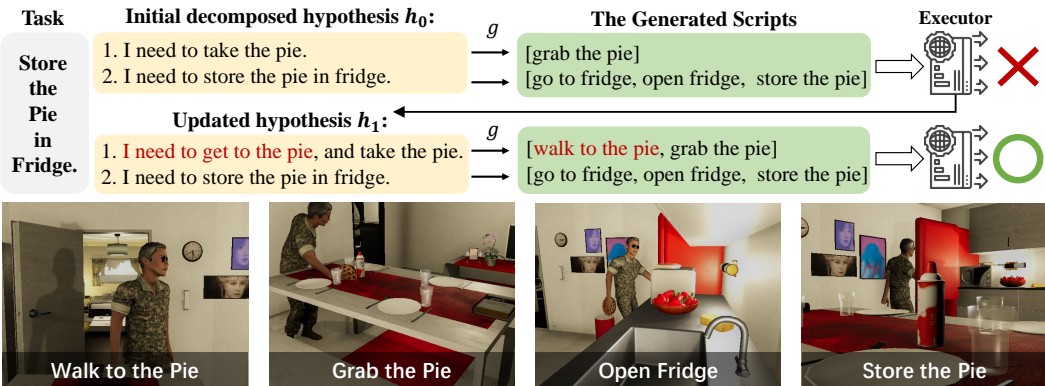

Figure 4: We demonstrate how RHDA can be extended to the VirtualHome framework to successfully complete the task of storing the pie in fridge.

- Difficulty in Generating Accurate Sub-Hypotheses: The generation of sub-hypotheses during the reasoning process often proves inaccurate, leading to subsequent breakdowns in reasoning chains.

- Sensitivity to Initial Hypotheses: The model exhibits a pronounced dependency on its initial hypotheses. Even when feedback is provided through amendment submissions, the model struggles to break free from its original thought framework, constraining its reasoning capabilities.

## 4.5 RHDA IS A FLEXIBLE AND SCALABLE PROBLEM-SOLVING PATHWAY

We consider extending the RHDA pipeline to more complex scenarios. To this end, we select VirtualHome (Puig et al., 2018; 2020), a sophisticated multi-agent platform for simulating household activities, as our new exploration subject. VirtualHome comprises a set of predefined atomic actions and objects that can be combined into high-level instructions. For example, '⟨char0⟩ [walk] ⟨salmon⟩' describes character 0 walking to the salmon. Given a specific scenario, the LLM is tasked with completing concrete housework using a series of high-level instructions. As depicted in Figure 4, and guided by the RHDA process, we demonstrate how the LLM successfully accomplishes the task of storing pie in the fridge through the methods of hypothesis decomposition, execution verification (offloading to VirtualHome engine), and reflection. we show another example in Appneidx D.

## 5 LIMITATION AND DISCUSSIONS

**Benchmark Selection.** This paper represents the first systematic exploration of the code reasoning task, focusing on the analysis of three forms of logical reasoning: inductive, deductive, and abductive. Due to time and cognitive constraints, we were unable to collect all benchmarks for testing. Our aim is to stimulate in-depth discussion on this topic and inspire meaningful follow-up research. While several excellent studies utilize code to address logical reasoning tasks (Zelikman et al., 2023; Hu et al., 2023; Srivastava et al., 2024; Liu et al., 2024a), we did not include them here due to their differing starting points from this paper.

**Hyperparameters.** The goal of this paper is to explore the potential of LLMs in code reasoning, rather than solely improving the performance of a specific code reasoning task. The RHDA framework serves as a preliminary exploration process; therefore, we didn't fully optimized the prompt templates or specific hyperparameters (such as temperature, $T$, and $N$) utilized. In the inductive code reasoning task, we examined a broader range of hyperparameter settings to illustrate that exploring multiple pathways aids in more effectively solving problems.

**Task Assessment.** We propose a novel code reasoning task, and experimental results indicate that current state-of-the-art LLMs exhibit limitations in tackling this task. In the future, we aim to further explore this challenging area and investigate the boundaries of human capabilities in similar tasks.

## 6 RELATED WORK

**Reasoning with LLMs.** LLMs such as GPT (OpenAI, 2023), LLaMA (Touvron et al., 2023), and Claude (Anthropic, 2024), demonstrate impressive reasoning capabilities across various NLP tasks (Zhang et al., 2024). However, due to the problems of direct reasoning with LLMs such as hallucinations (Ji et al., 2023), researchers have proposed several methods to enhance the reasoning power of LLMs. For example, (Zhou et al., 2023; Xue et al., 2025) decompose complex tasks into sequential subproblems, while (Sun et al., 2024) refine reasoning through environment feedback. Moreover, intermediate representations, such as graphs (Jiang et al., 2024), planning domain definition languages (PDDL) (Guan et al., 2023), and triples (Wang et al., 2023a), have been employed to enhance LLM's reasoning. Most recently, OpenAI o1 (OpenAI, 2024) demonstrates strong reasoning capabilities and broad world knowledge. Upon further contemplation, it is capable of reasoning through complex tasks and addressing challenges that exceed those faced by previous scientific, coding, and mathematical models.

Simultaneously, domain-specific reasoning with LLMs has gained attention. (Kim et al., 2024a) enhance reasoning outputs in computer tasks through recursive critique. In a case study using Minecraft, (Wang et al., 2023d) introduce a Describe, Interpret, Plan, and Select framework for open-world multitasking. In computer vision, (Gupta & Kembhavi, 2023) employ Python-like modular programs to tackle complex tasks. Nonetheless, reasoning in code remains an area yet to be thoroughly explored.

**Improvement with Reflection.** Reflective ability is regarded as a crucial metric for evaluating LLMs as agents. Reflection can be categorized into internal and external based on its feedback source (Pan et al., 2024). Internal reflection relies feedback from the model's own knowledge and parameters (Huang et al., 2022), while external feedback comes from various sources, including humans (Wang et al., 2023b), other models (Paul et al., 2024), external tools (Gou et al., 2024; Chen et al., 2024), or knowledge bases (Yao et al., 2023b; Asai et al., 2024). (Huang et al., 2024) find that LLMs struggle to self-correct their responses without external feedback, and in some cases, their performance may even decline following self-correction. Our work focuses on leveraging external tools, such as compilers, to generate feedback and enhance the performance of LLMs.

## 7 CONCLUSION

In this paper, we emphasized that the reasoning capabilities of LLMs still depend on recalling prior knowledge and highlighted that code reasoning has not been sufficiently explored as a novel perspective for examining the boundaries of LLM capabilities. Based on this consideration, we designed three meta-benchmarks—inductive code reasoning, deductive code reasoning, and abductive code reasoning—drawing on established forms of logical reasoning, and instantiated these benchmarks into eight specific tasks. Experimental results indicated that these benchmarks present significant challenges for current state-of-the-art LLMs. To initially explore code reasoning tasks, we proposed a method involving **R**eflective **H**ypothesis **D**ecomposition and **A**mendment (**RHDA**). This method was iterative: LLMs need to generate decomposed initial hypotheses based on observations and employ a translator to interpret these into executable functions that can be directly applied to the observations. After obtaining the executable functions, we performed execution verification and submit amendments, allowing for reflection and refinement of the sub-hypotheses. Experimental results demonstrated that this approach, which integrated the principles of divide-and-conquer and reflection, can flexibly solve complex code reasoning problems, achieving performance improvements of 2 to 3 times compared to baseline methods. Finally, we extended this process to simulate household tasks in real-world complex scenarios to validate its scalability and transferability.

## 8 ACKNOWLEDGMENT

This research was partially supported by the Key Technologies R&D Program of Anhui Province (No.202423k09020039), the National Natural Science Foundation of China (Grants No.62477044, 62406303), Anhui Provincial Natural Science Foundation (No. 2308085QF229), the Fundamental Research Funds for the Central Universities (No.WK2150110038, WK2150110034).

## 9 REPRODUCIBILITY STATEMENT

Our code, datasets and experimental results are available at `https://github.com/TnTWoW/code_reasoning`. Additionally, Appendix H contains details about pipeline and prompts used in method.

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

## A  DSL GRAMMARS

RobustFill is a string manipulation task using the DSL. Figure 5 illustrates the DSL syntax for RobustFill. Our implementation is based on the works of ExeDec (Shi et al., 2024) and Robust-Fill (Devlin et al., 2017).

Deepcoder is a list transformation task using the DSL. Figure 6. This implementation is based on the works of ExeDec (Shi et al., 2024) and DeepCoder (Balog et al., 2016).

$$
\begin{array}{rcl}
\text{Program } P & := & \texttt{Concat}(e_1, e_2, \ldots) \\
\text{Expression } e & := & s \mid m \mid o \mid \texttt{ConstStr}(c) \\
\text{Compose } o & := & m_1(m_2) \mid m(s) \\
\text{Substring } s & := & \texttt{SubStr}(k_1, k_2) \mid \texttt{GetSpan}(r_1, i_1, b_1, r_2, i_2, b_2) \mid \texttt{GetToken}(r, i) \\
& & \mid \texttt{GetUpto}(r) \mid \texttt{GetFrom}(r) \\
\text{Modification } m & := & \texttt{ToCase}(a) \mid \texttt{Replace}(\delta_1, \delta_2) \mid \texttt{Trim}() \mid \texttt{GetFirst}(r, i) \mid \texttt{GetAll}(r) \\
& & \mid \texttt{Substitute}(r, i, c) \mid \texttt{SubstituteAll}(r, c) \mid \texttt{Remove}(r, i) \mid \texttt{RemoveAll}(r) \\
\text{Regex } r & := & \texttt{NUMBER} \mid \texttt{WORD} \mid \texttt{ALPHANUM} \mid \texttt{ALL\_CAPS} \mid \texttt{PROPER\_CASE} \mid \texttt{LOWER} \mid \texttt{DIGIT} \mid \texttt{CHAR} \mid \delta \\
\text{Case } a & := & \texttt{ALL\_CAPS} \mid \texttt{PROPER\_CASE} \mid \texttt{LOWER} \\
\text{Position } k & := & -100 \mid -99 \mid \ldots \mid -1 \mid 0 \mid 1 \mid 2 \mid \ldots \mid 100 \\
\text{Index } i & := & -5 \mid -4 \mid \ldots \mid -1 \mid 1 \mid 2 \mid \ldots \mid 5 \\
\text{Boundary } b & := & \texttt{START} \mid \texttt{END} \\
\text{Character } c & := & A \mid \ldots \mid Z \mid a \mid \ldots \mid z \mid 0 \mid \ldots \mid 9 \mid \delta \\
\text{Delimiter } \delta & := & \texttt{\&,.?!@()[]\%\{\}/:;\$\#\ "'}
\end{array}
$$

Figure 5: The DSL syntax for string manipulation tasks in the RobustFill domain.

$$
\begin{array}{rcl}
\text{Program } P & := & i_1; i_2; \ldots; a_1; a_2; \ldots \\
\text{Initialization } i & := & v \leftarrow \texttt{INPUT} \\
\text{Assignment } a & := & v \leftarrow f \mid v \leftarrow h \\
\text{First-Order Operation } f & := & \texttt{Head}(l) \mid \texttt{Last}(l) \mid \texttt{Access}(n, l) \mid \texttt{Minimum}(l) \mid \texttt{Maximum}(l) \mid \texttt{Sum}(l) \\
& & \mid \texttt{Take}(n, l) \mid \texttt{Drop}(n, l) \mid \texttt{Reverse}(l) \mid \texttt{Sort}(l) \\
\text{Higher-Order Operation } h & := & \texttt{Map}(\lambda, l) \mid \texttt{Filter}(\beta, l) \mid \texttt{Count}(\beta, l) \mid \texttt{ZipWith}(\Sigma, l, l) \mid \texttt{Scanl1}(\Sigma, l) \\
\text{int} \rightarrow \text{int Lambda } \lambda & := & (+1) \mid (-1) \mid (*2) \mid (/2) \mid (*(-1)) \mid (**2) \mid (*3) \mid (/3) \mid (*4) \mid (/4) \\
\text{int} \rightarrow \text{bool Lambda } \beta & := & (>0) \mid (<0) \mid (\%2 == 0) \mid (\%2 == 1) \\
\text{(int, int)} \rightarrow \text{int Lambda } \Sigma & := & (+) \mid (-) \mid (*) \mid (\min) \mid (\max) \\
\text{Integer Variable } n & := & v \\
\text{List Variable } l & := & v \\
\text{Variable Name } v & := & x_1 \mid x_2 \mid \ldots
\end{array}
$$

Figure 6: The DSL for integer and list manipulation tasks in the DeepCoder domain.

## B  EXPERIMENTAL RESULTS USING MORE LLMs

We report the performance of Llama3.1-70B-Instruct, Qwen-max (qwen-max-2024-09-19), Claude 3.5 (claude-3-5-sonnet-20240620) using the RHDA method and compare them with GPT-4o (gpt-4o-2024-0806). The results for inductive code reasoning are shown in Table 5. The experimental results indicate that GPT-4o performs better in solving DSL problems, while Claude 3.5 excels in General Propose Language (GPL) tasks. Compared to closed-source models, the open-source model Llama still exhibits relatively limited reasoning capabilities. However, in list manipulation tasks (List Function and Deepcoder), Llama demonstrates stronger programming abilities. In Table 6, we

Table 5: Performance comparison of Llama3.1-70B-Instruct, Qwen-max, Claude 3.5 and GPT-4o on the PoT and RHDA methods in inductive code reasoning task. $T$ refers to the maximum number of iterations. $N$ refers to the number of candidates.

| Model | Method | Accuracy | | | | Task Accuracy | | | |
|---|---|---|---|---|---|---|---|---|---|
| | | MiniARC | List Func | RobustFill | DeepCoder | MiniARC | List Func | RobustFill | DeepCoder |
| Llama3.1 | PoT | 3.08 | 35.25 | 14.78 | 22.92 | 1.54 | 26.80 | 8.70 | 11.46 |
| | Sub-Hyp | 3.33 | 26.45 | 13.04 | 18.06 | 3.08 | 20.40 | 4.35 | 6.25 |
| | T=2, N=1 | 3.85 | 32.35 | 20.87 | 11.46 | 3.85 | 26.40 | 13.04 | 7.29 |
| Qwen-max | PoT | 6.41 | 41.75 | 36.52 | 25.35 | 3.85 | 30.00 | 21.74 | 14.58 |
| | Sub-Hyp | 5.90 | 46.25 | 26.09 | 17.36 | 3.08 | 36.40 | 8.70 | 5.21 |
| | T=2, N=1 | 6.41 | 46.60 | 33.91 | 24.64 | 3.08 | 41.60 | 13.04 | 10.42 |
| Claude-3.5 | PoT | 11.79 | 51.30 | 30.43 | 25.69 | 8.46 | 39.20 | 27.14 | 13.54 |
| | Sub-Hyp | 12.56 | 53.55 | 22.61 | 33.33 | 9.23 | 42.40 | 8.70 | 16.67 |
| | T=2, N=1 | **18.21** | **57.95** | 33.91 | 29.86 | **13.85** | **48.40** | 17.39 | 20.83 |
| GPT-4o | PoT | 10.90 | 44.90 | 37.39 | 30.90 | 8.46 | 33.60 | 26.09 | 19.79 |
| | Sub-Hyp | 8.46 | 47.10 | 35.65 | 24.65 | 6.92 | 36.40 | 17.39 | 12.50 |
| | T=2, N=1 | 12.56 | 51.05 | **43.48** | **38.89** | 10.77 | 41.20 | **40.43** | **23.96** |

Table 6: Performance comparison of Llama3.1-70B-Instruct, Qwen-max, Claude 3.5 and GPT-4o on the CoT and RHDA methods in deductive and abductive code reasoning tasks. $T$ refers to the maximum number of iterations. $N$ refers to the number of candidates.

| Model | Method | Deductive | | Abductive | |
|---|---|---|---|---|---|
| | | CRUXEval | LiveCodeBench | CRUXEval | LiveCodeBench |
| Llama3.1 | CoT | 40.25 | 7.84 | 53.12 | 38.24 |
| | Sub-Hyp | 30.75 | 6.86 | 50.88 | 8.82 |
| | T=2, N=1 | 45.62 | 10.78 | 59.62 | 40.20 |
| Qwen-max | CoT | 81.12 | 86.27 | 75.12 | 58.82 |
| | Sub-Hyp | 78.25 | 81.37 | 72.25 | 59.80 |
| | T=2, N=1 | 81.62 | **88.24** | 79.38 | 66.67 |
| Claude-3.5 | CoT | 82.75 | 77.45 | 73.62 | 61.76 |
| | Sub-Hyp | 77.75 | 65.69 | 74.75 | 53.92 |
| | T=2, N=1 | 86.88 | 80.39 | 83.38 | 61.76 |
| GPT-4o | CoT | 89.12 | 83.14 | 71.00 | 66.67 |
| | Sub-Hyp | 86.62 | 71.29 | 77.12 | 60.78 |
| | T=2, N=1 | **90.62** | 84.16 | **83.75** | **71.57** |

report the performance of the models in deductive and abductive code reasoning together. The experimental results show that GPT-4o outperforms Claude 3.5 in terms of program understanding and execution capabilities. These results suggest that RHDA is a framework-agnostic general process that can achieve optimal performance through a single reflection, applicable to both Llama, Qwen, Claude and GPT series models.

## C  BENCHMARK DETAILS

**List Function.** We use the original dataset (Rule, 2020), which consists of a total of 250 tasks. Due to the limited context lengths of LMs, we only use the first 16 examples from BIG-Bench (bench authors, 2023): 8 for seen examples and 8 for unseen examples. We manually examined the exemplars and found 8 examples are generally sufficient to describe the pattern.

**MiniARC.** We use the data from (Qiu et al., 2024). Such tasks are typically difficult to describe in natural language at an abstract level. Therefore,

Table 7: The number of tasks per dataset, the numbers of seen examples per task, and unseen examples per task.

| Dataset | # Tasks | # Seen | # Unseen |
|---|---|---|---|
| List Function | 250 | 8 | 8 |
| MiniARC | 130 | 3 | 3 |
| RobustFill | 22 | 5 | 5 |
| Deepcoder | 96 | 3 | 3 |
| CRUXEval | 800 | 1 | 1 |
| LiveCodeBench | 102 | 1 | 1 |

we did not consider them for our evaluations. As we only evaluate textonly models, we use textual representations of the original visual grids by mapping each cell to a corresponding integer.

**RobustFill.** RobustFill is a string manipulation task where the model is expected to perform a combination of atomic operations, such as extracting a substring from position $k_1$ to $k_2$ using $SubString(k_1, k_1)$, to achieve generalization. As an example, a program `ToCase(Lower, SubStr(1,3))` converts full month names (January, April) to their abbreviations (jan, apr).

**DeepCoder.** The DeepCoder task involves using DSL to perform operations on integer lists. In DeepCoder, each line represents a subroutine that performs atomic operations on previous variables and assigns the results to new variables. The result of the final line is the program's output. For example, program `a ← [int] | b ← FILTER(<0) a | c ← MAP(*4) b | d ← SORT c | e ← REVERSE b` (where "|" denotes subroutine separator.) transforms the input `[-17, -3, 4, 11, 0, -5, -9, 13, 6, 6, -8, 11]` into the output `[-12, -20, -32, -36, -68]`

# D   RHDA ACTING AS AN AGENT IN VIRTUALHOME

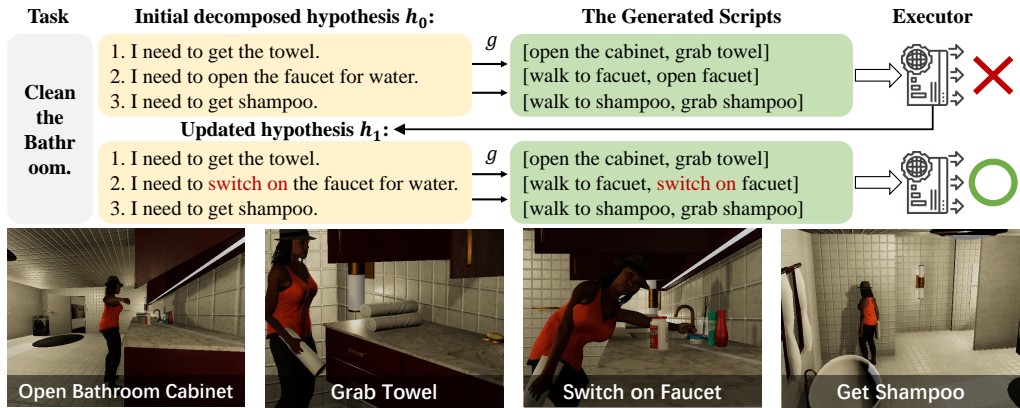

Figure 7: We illustrate how the RHDA framework can be extended to the VirtualHome environment to effectively accomplish the task of cleaning the bathroom.

We utilized the RHDA framework to drive agent actions in the VirtualHome environment powered by LLMs. Figure 7 illustrates a task of cleaning the bathroom.

Table 8: Execution Error Rate on VirtualHome

|  | native GPT-4o | w/o Sub-Hyp | w/o Amend | RHDA |
|---|---|---|---|---|
| # Error Action ↓ | 92 | 84 | 84 | **52** |
| Avg. Err per Step ↓ | 0.84 | 0.35 | 0.20 | **0.16** |
| Avg. Err per Task ↓ | 2.09 | 1.83 | 1.75 | **1.08** |

We also provided some quantitative metrics to validate the potential of RHDA as a agent in VirtualHome. Specifically, we selected a total of 52 tasks across two scenarios in VirtualHome and manually tested their execution error rates. The test results are shown in Table 8, which indicate that native GPT-4o struggles to handle simulated real-world scenarios effectively. The primary cause of failure lies in generating scripts that, while semantically similar to correct actions, are not executable within the environment (e.g., 'open the tap' is invalid action, whereas 'touch the tap' is valid action). By employing the RHDA method, which incorporates step-by-step solutions and effective feedback mechanisms, the error rate was significantly reduced.

# E EXAMPLES ANALYSES

## E.1 EFFECTIVE CASE STUDY

Table 9: We compare the results obtained using the sub-hypothesis decomposition method with those obtained without it. The results without hypothesis decomposition are presented at the top of the table, while those with hypothesis decomposition are shown below. Benchmark ARC-ID37.

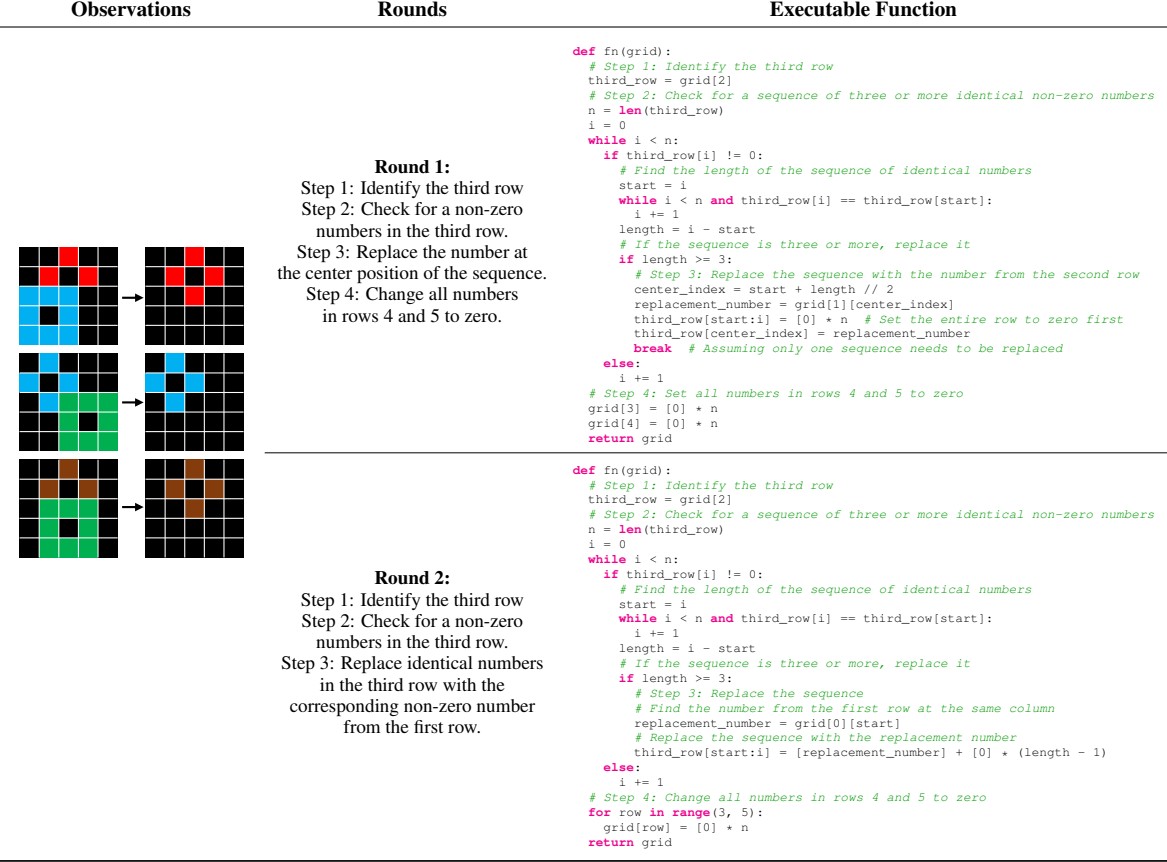

| Observations | Rounds | Executable Function |
|---|---|---|
| | **Round 1:**
Step 1: Identify the third row
Step 2: Check for a non-zero
numbers in the third row.
Step 3: Replace the number at
the center position of the sequence.
Step 4: Change all numbers
in rows 4 and 5 to zero. | ```python
def fn(grid):
    # Step 1: Identify the third row
    third_row = grid[2]
    # Step 2: Check for a sequence of three or more identical non-zero numbers
    n = len(third_row)
    i = 0
    while i < n:
        if third_row[i] != 0:
            # Find the length of the sequence of identical numbers
            start = i
            while i < n and third_row[i] == third_row[start]:
                i += 1
            length = i - start
            # If the sequence is three or more, replace it
            if length >= 3:
                # Step 3: Replace the sequence with the number from the second row
                center_index = start + length // 2
                replacement_number = grid[1][center_index]
                third_row[start:i] = [0] * n   # Set the entire row to zero first
                third_row[center_index] = replacement_number
                break   # Assuming only one sequence needs to be replaced
        else:
            i += 1
    # Step 4: Set all numbers in rows 4 and 5 to zero
    grid[3] = [0] * n
    grid[4] = [0] * n
    return grid
``` |
| | **Round 2:**
Step 1: Identify the third row
Step 2: Check for a non-zero
numbers in the third row.
Step 3: Replace identical numbers
in the third row with the
corresponding non-zero number
from the first row. | ```python
def fn(grid):
    # Step 1: Identify the third row
    third_row = grid[2]
    # Step 2: Check for a sequence of three or more identical non-zero numbers
    n = len(third_row)
    i = 0
    while i < n:
        if third_row[i] != 0:
            # Find the length of the sequence of identical numbers
            start = i
            while i < n and third_row[i] == third_row[start]:
                i += 1
            length = i - start
            # If the sequence is three or more, replace it
            if length >= 3:
                # Step 3: Replace the sequence
                # Find the number from the first row at the same column
                replacement_number = grid[0][start]
                # Replace the sequence with the replacement number
                third_row[start:i] = [replacement_number] + [0] * (length - 1)
        else:
            i += 1
    # Step 4: Change all numbers in rows 4 and 5 to zero
    for row in range(3, 5):
        grid[row] = [0] * n
    return grid
``` |

Table 10: We compare the results obtained using the sub-hypothesis decomposition method with those obtained without it. The results without hypothesis decomposition are presented at the top of the table, while those with hypothesis decomposition are shown below. Benchmark List Function-ID2.

| Observations | Hypothesis | Executable Function |
|---|---|---|
| [] → []
[6, 9] → []
[1, 5, 0, 6, 2, 9, 3] → [0]
[6, 3, 4, 1, 7, 2, 9, 8, 0] → [4]
... | **No Sub Hypothesis:**
The output is the element
from the input list that
is exactly in the middle
of the list. | ```python
def fn(input_list):
    # Check if the list is empty or has only one element
    if len(input_list) <= 1:
        return []
    # Calculate the middle index
    middle_index = (len(input_list) - 1) // 2
    # Return the middle element as a list
    return [input_list[middle_index]]
``` |
| | **Sub Hypothesis:**
Step 1: If the list has fewer
than three elements,
the output is an empty list.
Step 2:If the list has three
or more elements,
the output is a list containing
only the integer at the third
position. | ```python
def fn(input_list):
    # Check if the list has fewer than three elements
    if len(input_list) < 3:
        return []
    else:
        # Return a list containing the integer
        # at the third position (index 2)
        return [input_list[2]]
``` |

We validated the effectiveness of the proposed method using examples from various benchmarks. For instance, as shown in Table 9, the MiniARC task example with ID 37 highlights how the LLM, after receiving feedback, successfully reflects on its errors and submits a revised solution.

Table 11: We compare the results obtained using the sub-hypothesis decomposition method with those obtained without it. The results without hypothesis decomposition are presented at the top of the table, while those with hypothesis decomposition are shown below. Benchmark Livecodebench Input-ID37.

| Observations | Hypothesis | Executable Function |
|---|---|---|
| ```
def minOperations(a, b):
  def f():
    ret = 0
    aa, bb = a[:], b[:]
    for i in range(len(a)):
      if a[i] > a[-1] or b[i] > b[-1]:
        a[i], b[i] = b[i], a[i]
        ret += 1
      if a[i] > a[-1] or b[i] > b[-1]:
        return inf
    a[:] = aa
    b[:] = bb
    return ret

  ans = f()
  a[-1], b[-1] = b[-1], a[-1]
  ans = min(ans, f() + 1)
  return -1 if ans > len(a) else ans
  # assert f(??) == 1
``` | **No Sub Hypothesis:**
The function 'minOperations' attempts to determine the minimum number of swaps needed to ensure that for each index 'i', neither 'a[i]' nor 'b[i]' are greater than the last elements of their respective lists. | assert minOperations([3, 1, 2], [1, 3, 2]) == 1 |
| | **Sub Hypothesis:**
Step 1: **Function Purpose**
The goal of 'minOperations' is to ...
Step 2: **Inner Function**
It attempts to ensure that for each 'i', ...
Step 3: **Main Logic**
It calculates 'ans' by calling 'f', ...
Step 4: **Objective**
To find an input such that... | assert minOperations([2, 3], [3, 2]) == 1 |

Table 12: We present the additional improvements resulting from the use of amendments. The results shown above are those obtained after the initial hypotheses, while the results displayed below reflect the outcomes following the submission of amendments and subsequent reflections. Benchmark CruxEval Output-ID328.

| Observations | Rounds | Executable Function |
|---|---|---|
| ```
def f(array, L):
  if L <= 0:
    return array
  if len(array) < L:
    array.extend(f(array, L - len(array)))
  return array
  # assert f([1, 2, 3], 4) == ??
``` | **Round 1:**
Step 1 Base Case Check:
The function checks if 'L' is less than or equal to 0.
If true, it simply returns the array as is
Step 2 Recursive Extension:
If the length of 'array' is less than L, it then calls itself recursively
Step 3 Return Array:
If the array is already of length 'L' or longer, the function simply returns the array without any modifications. | assert f([1, 2, 3], 4) == [1, 2, 3, 1] |
| | **Round 2:**
Step 1 Base Case Check: ...
Step 2 Recursive Extension: ...
Step 3 Return Array: ...
Step 4 Example Check:
The original call then extends '[1, 2, 3]' by '[1, 2, 3]', resulting in '[1, 2, 3, 1, 2, 3]'. | assert f([1, 2, 3], 4) == [1, 2, 3, 1, 2, 3] |

In Table 10, hypothesis decomposition reveals that the output number is determined not only by its position at the middle of the input array but also by being the third character.

In Table 11, compared to models without hypothesis decomposition, those utilizing this approach progressively analyze the function's behavior, ultimately achieving an abstract understanding of the program and making accurate assertions. In Table 12, for a complex recursive function, while the LLM accurately grasped the overall functionality of the function through hypothesis decomposition, it encountered difficulties during the detailed analysis of specific instances. Following the submission of a revised solution, the LLM reflected on its errors and successfully resolved the issue, addressing the collapse of the overall logical chain caused by a failure in a single step during multi-step reasoning.

Table 13: Failure case study. The observations indicate that the hypotheses proposed by the LLM are overly simplistic, rendering it difficult to account for all possible cases. Additionally, manual inspection and efforts to summarize these hypotheses proved challenging. This limitation is one of the primary factors contributing to the LLM's failure in this task. Benchmark: List Function-ID9.

| Observations | Rounds | Executable Function |
|---|---|---|
| [2, 8] → [8]
[7, 5, ..., 8, 4] → [5, ..., 8]
[8, 2, ..., 9] → [2, ..., 9]
[3, 2, 1, 0, 7, 8] → [2, 1, 0]
...
... | **Round 1:**
Remove the first element from the input list. | ```python
def fn(input_list):
    """
    Removes the first element from the
    input list of integers.
    """
    # Check if the input list is empty
    if not input_list:
        return []

    # Return the list excluding the
    # first element
    return input_list[1:]
``` |
| | **Round 2:**
Remove the first and the last elements from the input list. | ```python
def fn(input_list):
    """
    Remove the first and last elements
    from the input list.
    """
    # Check if the list has less than
    # 2 elements
    if len(input_list) < 2:
        return []
    # Return the list excluding the
    # first and last elements
    return input_list[1:-1]
``` |

Table 14: Failure case study. The LLM make some redundant hypotheses, which led to failure. Even after amendments, it was unable to break free from its original thought framework, resulting in a failure to reflect effectively. Benchmark: ARC-ID5.

| Observations | Rounds | Executable Function |
|---|---|---|
| 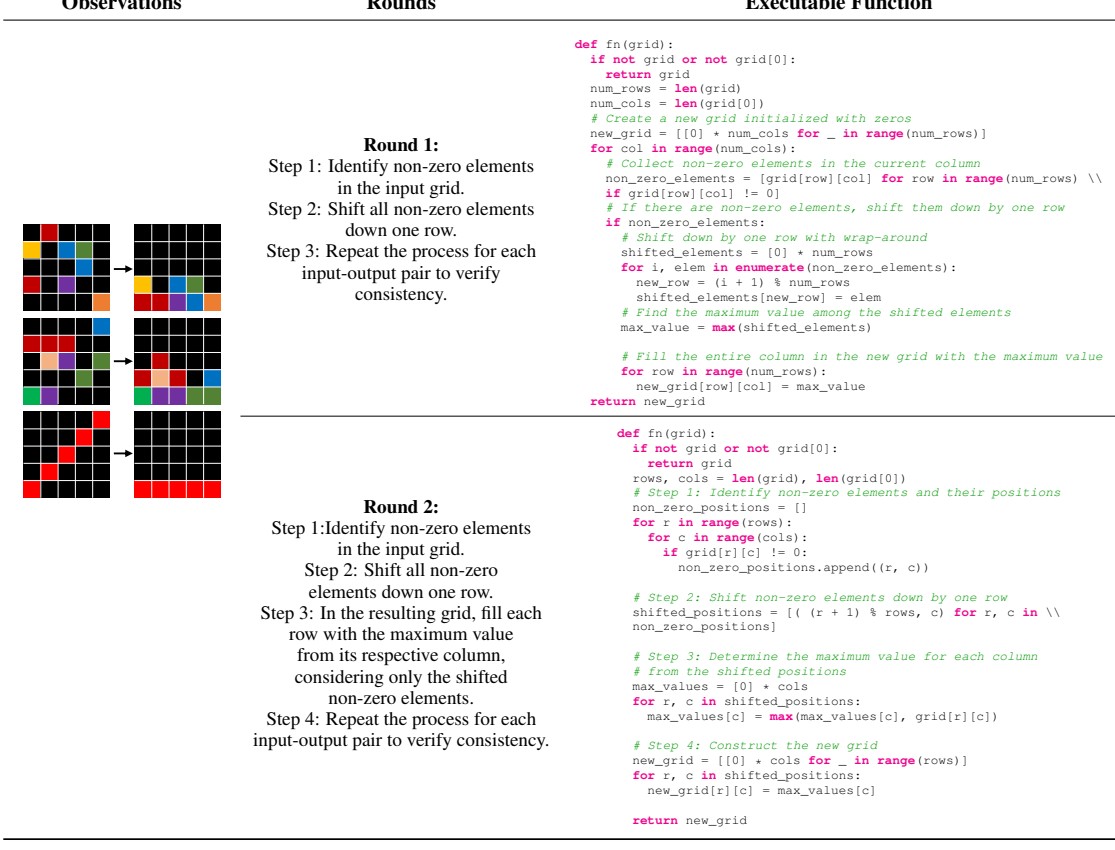 | **Round 1:**
Step 1: Identify non-zero elements in the input grid.
Step 2: Shift all non-zero elements down one row.
Step 3: Repeat the process for each input-output pair to verify consistency. | ```python
def fn(grid):
    if not grid or not grid[0]:
        return grid
    num_rows = len(grid)
    num_cols = len(grid[0])
    # Create a new grid initialized with zeros
    new_grid = [[0] * num_cols for _ in range(num_rows)]
    for col in range(num_cols):
        # Collect non-zero elements in the current column
        non_zero_elements = [grid[row][col] for row in range(num_rows) \\
        if grid[row][col] != 0]
        # If there are non-zero elements, shift them down by one row
        if non_zero_elements:
            # Shift down by one row with wrap-around
            shifted_elements = [0] * num_rows
            for i, elem in enumerate(non_zero_elements):
                new_row = (i + 1) % num_rows
                shifted_elements[new_row] = elem
            # Find the maximum value among the shifted elements
            max_value = max(shifted_elements)

            # Fill the entire column in the new grid with the maximum value
            for row in range(num_rows):
                new_grid[row][col] = max_value
    return new_grid
``` |
| | **Round 2:**
Step 1:Identify non-zero elements in the input grid.
Step 2: Shift all non-zero elements down one row.
Step 3: In the resulting grid, fill each row with the maximum value from its respective column, considering only the shifted non-zero elements.
Step 4: Repeat the process for each input-output pair to verify consistency. | ```python
def fn(grid):
    if not grid or not grid[0]:
        return grid
    rows, cols = len(grid), len(grid[0])
    # Step 1: Identify non-zero elements and their positions
    non_zero_positions = []
    for r in range(rows):
        for c in range(cols):
            if grid[r][c] != 0:
                non_zero_positions.append((r, c))

    # Step 2: Shift non-zero elements down by one row
    shifted_positions = [( (r + 1) % rows, c) for r, c in \\
    non_zero_positions]

    # Step 3: Determine the maximum value for each column
    # from the shifted positions
    max_values = [0] * cols
    for r, c in shifted_positions:
        max_values[c] = max(max_values[c], grid[r][c])

    # Step 4: Construct the new grid
    new_grid = [[0] * cols for _ in range(rows)]
    for r, c in shifted_positions:
        new_grid[r][c] = max_values[c]

    return new_grid
``` |

## E.2    FAILURE ANALYZE

We analyze RHDA's performance in numerous failure cases and summarize the underlying causes of these failures. Our findings suggest that the primary reason can be attributed to the insufficient

intrinsic capability of LLMs in code reasoning tasks. This limitation is specifically reflected in two aspects:

- **Sub-hypotheses fail to address the problem**: For tasks that are overly complex or abstract (e.g., cases shown in Table 13), even though hypothesis decomposition attempts to reduce task complexity, LLMs still struggle to handle them effectively.
- **Amendments fail to correct sub-hypotheses**: While amendments leverage external feedback to help LLMs reflect on their mistakes, the models often remain confined to their existing thought framework, even after recognizing errors (e.g., cases shown in Table 14). This results in the correction failing to resolve the issue.

These observations indicate that for tasks exceeding the intrinsic capabilities of LLMs, relying solely on reflective hypothesis decomposition and amendment may not be sufficient to improve the model's performance.

# F  COSTS

Table 15: Avg. API calls and Total Cost using GPT-4o.

| Method | Avg. API Calls | | | | Total Cost (cent) | | | |
|---|---|---|---|---|---|---|---|---|
| | List Func | MiniARC | RobustFill | Deepcoder | List Func | MiniARC | RobustFill | Deepcoder |
| IO | 8.0 | 4.0 | 5.0 | 3.0 | 10.2 | 4.6 | 2.0 | 3.3 |
| PoT | 1.0 | 1.0 | 1.0 | 1.0 | 5.0 | 3.7 | 0.6 | 1.2 |
| CoC | 1.0 | 1.0 | 1.0 | 1.0 | 11.0 | 9.0 | 1.1 | 1.4 |
| SC (N=3) | 3.0 | 24.0 | 15.0 | 9.0 | 5.3 | 3.7 | 0.6 | 1.2 |
| SR (T=2) | 1.4 | 1.9 | 1.5 | 1.6 | 4.6 | 3.3 | 0.5 | 1.1 |
| T=2, N=3 | 5.4 | 5.9 | 5.5 | 5.6 | 8.6 | 4.0 | 3.1 | 4.7 |

| Method | Deductive | | Abductive | | Deductive | | Abductive | |
|---|---|---|---|---|---|---|---|---|
| | CRUXEval | LiveCodeBench | CRUXEval | LiveCodeBench | CRUXEval | LiveCodeBench | CRUXEval | LiveCodeBench |
| Standard | 1.0 | 1.0 | 1.0 | 1.0 | 2.9 | 0.5 | 3.1 | 1.5 |
| CoT | 1.0 | 1.0 | 1.0 | 1.0 | 19.4 | 3.7 | 19.5 | 3.3 |
| SC (N=3) | 3.0 | 3.0 | 3.0 | 3.0 | 2.9 | 0.5 | 3.3 | 1.5 |
| SR (T=2) | 1.6 | 1.7 | 1.4 | 1.5 | 3.8 | 0.6 | 3.4 | 1.6 |
| CoC | 1.0 | 1.0 | 1.0 | 1.0 | 18.3 | 4.1 | 19.0 | 3.4 |
| T=2, N=1 | 1.6 | 1.7 | 1.4 | 1.5 | 19.0 | 4.4 | 18.8 | 4.4 |

In Table 15, we present the average number of API calls and the total cost for each task. We used GPT-4o, with an input cost of $0.0025/1K tokens and an output cost of $0.01/1K tokens. The results indicate that our approach still demonstrates high cost-effectiveness for certain tasks.

# G  TRADE OFF BETWEEN NUMBER OF ITERATIONS AND PERFORMANCE GAIN

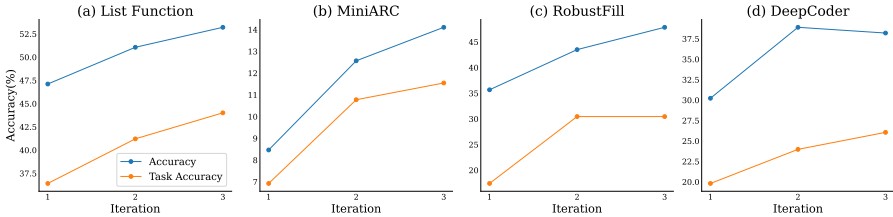

Figure 8: In the inductive code reasoning tasks, as the number of iterations increased, the performance continued to improve.

In this section, we investigate the impact of iteration count on the performance of three types of reasoning tasks, with experimental results illustrated in Figure 8 and Figure 9. For inductive and abductive code reasoning tasks, performance consistently improved as the number of iterations increased. However, the rate of improvement diminished, with marginal gains becoming less significant at higher iteration counts. Conversely, for deductive code reasoning tasks, performance followed a rise-and-fall trend, initially improving but declining with excessive iterations. These findings suggest that while increasing the number of iterations can enhance performance for general code reasoning tasks, it is crucial to balance iterative gains against potential performance instability.

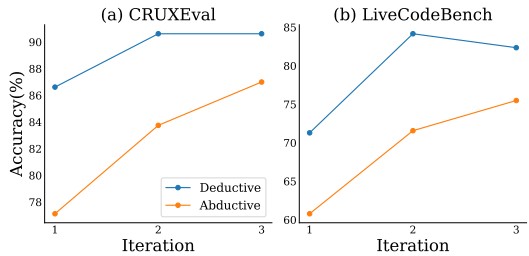

Figure 9: In the deductive code reasoning tasks, the performance slightly decreased as the number of iterations increased. Conversely, in the abductive code reasoning tasks, the performance consistently improved with an increasing number of iterations.

# H   PROMPTS

Table 16: Prompts used in our study. {} refers to a placeholder.

| Type | Prompt |
|---|---|
| Sub Hypothesis Generation | Generate a rule that maps the following inputs to their corresponding outputs step by steps. **{Task description}**

**{Examples}**

Please format your rule as follows:

**{Rule format}** |
| Amendment Submission | Your rule: **{Rule}**

This rule does not work for the following examples.

**{Feedback}**

Please carefully reconsider each of your steps to ensure that the rules are correct. Systematically generate new rules, step by step. **{Feedback description}** Please format your rule as follows:

**{Rule format}** |
| Hypothesis Translation | You are an expert Python programmer. Write a Python function `fn` for the following rule. **{Translation Example description}**

Rule: **{Rule}** |
| Rule Application | Generate an output corresponding to the given input based on the rule. **{Application Example description}**

Rule: **{Rule}**

Input: **{Test input}**
Output: |

