# OpenReview forum: "Unveiling the Magic of Code Reasoning through Hypothesis Decomposition and Amendment"
_ICLR.cc/2025/Conference — ICLR 2025 Poster_

### Official Review · Reviewer_JURK · 2024-10-17

**Soundness:** 2
**Presentation:** 3
**Contribution:** 3
**Rating:** 6
**Confidence:** 4

**Summary:**

This paper evaluates LLMs on code reasoning tasks, which require both recall and reasoning capabilities. The authors also propose a method RDHA which generally improves their performance in these tasks.

**Strengths:**

originality: RDHA is a novel method. It is interesting to see such a method to be applied to code reasoning and even planning tasks (e.g., Virtual Home).

quality: the paper conducts a systematic investigation of code reasoning tasks and an additional planning task, which is a very comprehensive task setting.

clarity: the paper is generally easy to follow.

significance: it is important to understand LLMs' capabilities and limitations.

**Weaknesses:**

Generally, the contributions of this paper are not strong enough.

1. If the main contribution is the evaluation of LLMs on code reasoning, it is not enough to test only two LLMs. Plus, it is not surprising that LLMs fail in complex reasoning tasks as widely testified in many existing works ([1,2], inter alia).

2. If the strongest contribution is RHDA, I wonder how it compares to other prompting methods such as self-refine [3].

2. Some arguments are a bit loose or controversial. For instance, the authors "...position the code reasoning task between memory and reasoning" because it requires memory of syntax from pre-training and reasoning about current question and context. In this sense, most, if not all of the reasoning tasks can be put in the middle of the Figure 1 spectrum, which has one extreme of reasoning and another extreme of recall. For instance, math, which the authors put on the reasoning extreme of the spectrum, requires the LLM to understand what operators mean as well (i.e., syntax, which also needs to come from training memory). The authors are encouraged to be more cautious in statements.

[1] Zhuo, T. Y., Vu, M. C., Chim, J., Hu, H., Yu, W., Widyasari, R., ... & Von Werra, L. (2024). Bigcodebench: Benchmarking code generation with diverse function calls and complex instructions. arXiv preprint arXiv:2406.15877.

[2] Dziri, N., Lu, X., Sclar, M., Li, X. L., Jiang, L., Lin, B. Y., ... & Choi, Y. (2024). Faith and fate: Limits of transformers on compositionality. Advances in Neural Information Processing Systems, 36.

[3] Madaan, A., Tandon, N., Gupta, P., Hallinan, S., Gao, L., Wiegreffe, S., ... & Clark, P. (2024). Self-refine: Iterative refinement with self-feedback. Advances in Neural Information Processing Systems, 36.

**Questions:**

1. What is the difference between Sub-Hyp and Chain of Thought?

2. I didn't get the rationale of why the IO prompt is the best in terms of accuracy. More specifically, what does it mean when you say this? On page 6, the end of the paragraph preceding the ablation study says "This may be a misunderstanding... than RHDA."?

3. At the bottom of page 7, "...we present a quantitative...", did you mean qualitative?

4. Did you conduct a full-scale experiment on Virtual Home? It seems that you only present two examples in the paper, which I'm not sure can generalize to the conclusion that RHDA is scalable and flexible.

5. What is the cost of RHDA? Does it generalize to other non-symbolic domains?

6. How does your method compare to self-refine?

---

> ### Author Response · Authors · 2024-11-22
> **Official Response to Reviewer JURK part (1/4)**
>
> We are very grateful for the Reviewer JURK's constructive suggestions. The following is our response:
>
> > If the main contribution is the evaluation of LLMs on code reasoning, it is not enough to test only two LLMs.
>
> We conducted additional experiments using two state-of-the-art models, **Llama3.1-70B-Instruct** and **Qwen-max**, to further reveal the limitations of LLMs on code reasoning tasks and to validate the effectiveness of the RHDA method. The experimental results are presented below.
>
> |                       |          | Accuracy |           |            |           | Task Accuracy |           |            |           |
> | --------------------- | -------- | :------: | :-------: | :--------: | :-------: | :-----------: | :-------: | :--------: | :-------: |
> | Model                 | Method   | MiniARC  | List Func | RobustFill | DeepCoder |    MiniARC    | List Func | RobustFill | DeepCoder |
> | Llama3.1-70B-Instruct | PoT      |   3.08   |   35.25   |   14.78    |   22.92   |     1.54      |   26.80   |    8.70    |   11.46   |
> |                       | Sub-Hyp  |   3.33   |   26.45   |   13.04    |   18.06   |     3.08      |   20.40   |    4.35    |   6.25    |
> |                       | T=2, N=1 |   3.85   |   32.35   |   20.87    |   11.46   |     3.85      |   26.40   |   13.04    |   7.29    |
> | Qwen-max              | PoT      |   6.41   |   41.75   |   36.52    |   25.35   |     3.85      |   30.00   |   21.74    |   14.58   |
> |                       | Sub-Hyp  |   5.90   |   46.25   |   26.09    |   17.36   |     3.08      |   36.40   |    8.70    |   5.21    |
> |                       | T=2, N=1 |   6.41   |   46.60   |   33.91    |   24.64   |     3.08      |   41.60   |   13.04    |   10.42   |
>
> |                       |          | Deductive |               | Abductive |               |
> | --------------------- | -------- | :-------: | :-----------: | :-------: | :-----------: |
> | Model                 | Method   | CRUXEval  | LiveCodeBench | CRUXEval  | LiveCodeBench |
> | Llama3.1-70B-Instruct | CoT      |   40.25   |     7.84      |   53.12   |     38.24     |
> |                       | Sub-Hyp  |   30.75   |     6.86      |   50.88   |     8.82      |
> |                       | T=2, N=1 |   45.62   |     10.78     |   59.62   |     40.20     |
> | Qwen-max              | CoT      |   81.12   |     86.27     |   75.12   |     58.82     |
> |                       | Sub-Hyp  |   78.25   |     81.37     |   72.25   |     59.80     |
> |                       | T=2, N=1 |   81.62   |     88.24     |   79.38   |     66.67     |
>
> All relevant results have been incorporated into the Appendix B in latest version of the manuscript. Furthermore, we have updated the running code and execution processes in our anonymous GitHub repository to ensure reproducibility.
>
> > it is not surprising that LLMs fail in complex reasoning tasks as widely testified in many existing works.
>
> We agree with the perspective that the shortcomings of current LLMs in reasoning tasks are unsurprising. Additionally, we remain cautious about whether LLMs genuinely possess reasoning capabilities. In this paper, our objective is to illustrate that, by leveraging the high-quality code knowledge acquired during pretraining, LLMs can achieve reasoning to a limited extent (where this extent depends on how reasoning problems are defined). To this end, we introduce the concept of **code reasoning** tasks, which seek to address some specific reasoning challenges through code-formed solutions. We hope this work will inspire the research community and foster further exploration in the domain of code reasoning.
>
> > I wonder how it compares to other prompting methods such as self-refine
>
> We first make a clear distinction between self-refine and external feedback-based refine. Although Madaan et al. (2023) [1] refers to their work as self-refine, the method actually relies on effective external feedback, which distinguishes it from true self-correct [2] processes. We report the self-refine technique as follows:
>
> |             | Accuracy |           |            |           | Task Accuracy |           |            |           |
> | ----------- | :------: | :-------: | :--------: | :-------: | :-----------: | :-------: | :--------: | :-------: |
> | Method      | MiniARC  | List Func | RobustFill | DeepCoder |    MiniARC    | List Func | RobustFill | DeepCoder |
> | PoT         |  10.90   |   44.90   |   46.09    |   30.90   |     8.46      |   33.60   |   26.09    |   19.79   |
> | Self Refine |  10.26   |   51.10   |   41.74    |   36.81   |     8.46      |   41.60   |   21.74    |   25.00   |
> | RHDA        |  19.74   |   58.35   |   54.78    |   43.06   |     13.85     |   48.80   |   34.78    |   29.13   |

---

> ### Author Response · Authors · 2024-11-22
> **Official Response to Reviewer JURK part (2/4)**
>
> |             | Deductive |               |                 | Abductive |               |
> | :---------: | :-------: | :-----------: | :-------------: | :-------: | :-----------: |
> |             | cruxeval  | livecodebench |                 | cruxeval  | livecodebench |
> |  Standard   |   68.75   |     41.18     |    Standard     |   63.50   |     50.98     |
> | Self Refine |   80.38   |     63.73     | Self Reflection |   74.50   |     65.69     |
> |    RHDA     |   90.62   |     84.16     |      RHDA       |   83.75   |     71.57     |
>
> Our conclusions align with [1, 2], emphasizing methods relying solely on self-correct often struggle to achieve optimal performance. However, when combined with external interpreters (e.g., compilers), iterative performance improves significantly. RHDA also relies on the provision of effective external information, facilitating performance improvements through a continuous iterative process. We have incorporated [1] as a baseline method into Tables 1 and 2, as well as Figure 1, in the revised version of the paper.
>
> > Some arguments are a bit loose or controversial. For instance, the authors "...position the code reasoning task between memory and reasoning" because it requires memory of syntax from pre-training and reasoning about current question and context. In this sense, most, if not all of the reasoning tasks can be put in the middle of the Figure 1 spectrum, which has one extreme of reasoning and another extreme of recall. For instance, math, which the authors put on the reasoning extreme of the spectrum, requires the LLM to understand what operators mean as well (i.e., syntax, which also needs to come from training memory).
>
> We appreciate the reviewer for highlighting this point. In fact, we carefully considered this question before writing the paper. In Figure 1 of the original manuscript, we did not place mathematical reasoning at the extreme end of the reasoning spectrum; instead, we appropriately positioned it closer to the middle. This decision aligns with the reviewer’s observation that mathematical reasoning also relies on knowledge acquired during the pretraining phase. Similarly, we did not place translation tasks entirely at the recall extreme, as the translation process also requires recall and analogy with previously learned knowledge (though limited reasoning is involved, as it is not feasible to train LLMs with all translation samples). We positioned code reasoning tasks in the middle, as programming languages adhere to strict syntactic rules while also exhibiting some inherent reasoning potential. The code reasoning task requires less leap in reasoning compared to mathematical reasoning and less reliance on memory recall compared to translation. This allows code reasoning to act as a bridge for "memory-based reasoning."
>
> > What is the difference between Sub-Hyp and Chain of Thought
>
> The primary difference between the two methods lies in the number of context examples provided. For **CoT**, a 2-shot setting is employed for deductive and abductive reasoning tasks, while **Sub-Hyp** utilizes a zero-shot setting. In theory, CoT is expected to outperform Sub-Hyp due to its higher informational input overhead. For the reviewers' convenience, the prompt templates used on LiveCodeBench Input prediction are provided below.
>
> ````CoT
> You will be given a function {func_name} and an output in the form {func_name}(??) == output.
> Your task is to find any input such that executing {func_name} on the input leads to the given output. There may be
> multiple answers, but only output one. First, think step by step. Express your answer as a passing assertion
> containing the input and the given output.
>
> ```python
> def f(x):
>     return x + 1
> # assert f(??) == 17
> ```
> Let's analyze the code step by step:
> To find an input such that executing f on the input leads to the given output, we can work backwards from the given assertion.
> We know that f(??) == 17. Since the function f(x) returns x + 1, for f(??) to be equal to 17, the value of ?? should be 16.
>
> Answer:
> ```python
> assert f(16) == 17
> ```
>
> ```python
> def f(x):
>     s = x + x
>     result = "b" + s + "a"
>     return result
> # assert f(??) == "bhihia"
> ```
> Let's analyze the code step by step:
> To find an input such that executing f on the input leads to the given output, we need to understand the function. This function repeat the input x twice and add a char 'b' and 'a' at the beginning and end of the repeated string. The output should remove 'b' and 'a', then remove the repeated string 'hi', resulting the original input 'hi'.
>
> Answer:
> ```python
> assert f("hi") == "bhihia"
> ```
>
> ```{language}
> {code}
> # assert {func_name}(??) == {output}
> ```
> Let's analyze the code step by step:
> ````
>
> And the following is prompt used in Sub-Hyp.

---

> ### Author Response · Authors · 2024-11-22
> **Official Response to Reviewer JURK part (3/4)**
>
> ````Sub-Hyp
> You will be given a function {func_name} and an output in the form {func_name}(??) == output.
> Your task is to find any input such that executing {func_name} on the input leads to the given output. There may be
> multiple answers, but only output one. First, analysis the program step by step. Express your answer as a passing
> assertion containing the input and the given output.
>
> ```{language}
> {code}
> # assert f(??) == {output}
> ```
>
> Please provide the analysis and the assertion in the following format:
> Analysis: <Your analysis>
> Answer:
> ```{language}
> assert {func_name}(<Your input>) == {output}
> ```
> ````
>
>
>
> > I didn't get the rationale of why the IO prompt is the best in terms of accuracy.
>
> List function have 8 seen examples and 8 unseen examples, predictions for the unseen examples must be conducted independently, necessitating 8 separate API calls. Each call includes the 8 seen examples and 1 unseen example as input. We also experimented with a single API call to infer all unseen examples simultaneously. However, this approach failed due to the high complexity of the task.  Below, we provide the I/O prompt to assist reviewers in understanding the process.
>
> ```IO prompt
> Generate an output corresponding to the given input. Each output is generated by applying the same function to the respective inputs.
>
> {seen examples}
>
> Input: {test_input}
> Output:
> ```
>
>
>
> > At the bottom of page 7, "...we present a quantitative...", did you mean qualitative?
>
> We sincerely thank the Reviewer JURK for diligent review. The identified typo has been corrected, and we have conducted a comprehensive review of the entire manuscript. The relevant revisions have been incorporated into the latest version of the paper.
>
> > Did you conduct a full-scale experiment on Virtual Home?
>
> VirtualHome is not a standardized benchmark dataset but an instruction-driven agent platform, and as such, it lacks both comprehensive standardized data and effective automated validation schemes for experimentation. To assess performance, we selected 52 tasks across two scenarios and **manually** verified their failure rates. The experimental results are presented in the table. Our findings indicate that the native GPT-4o faces significant difficulties in adapting to VirtualHome. The primary issue arises from its generated scripts being semantically similar but unrecognizable by VirtualHome (e.g., `open the tap` is invalid, while `touch the tap` is valid). In contrast, the RHDA method significantly reduces the error rate by employing step-by-step solutions and effective feedback.
>
> |                                  | native GPT-4o | w/o Sub-Hyp | w/o Amend | RHDA     |
> | -------------------------------- | ------------- | ----------- | --------- | -------- |
> | # Error Action $\downarrow$      | 92            | 84          | 84        | **52**   |
> | Avg. Error per Step $\downarrow$ | 0.84          | 0.35        | 0.20      | **0.16** |
> | Avg. Err per Task $\downarrow$   | 2.09          | 1.83        | 1.75      | **1.08** |

---

> ### Author Response · Authors · 2024-11-22
> **Official Response to Reviewer JURK part (4/4)**
>
> > What is the cost of RHDA? Does it generalize to other non-symbolic domains?
>
> We calculated the average number of API calls and the total cost associated with using GPT-4o as follows:
>
> **Avg. API Calls**
>
> |          | List Func | MiniARC | RobustFill | Deepcoder |
> | -------- | :-------: | :-----: | :--------: | :-------: |
> | IO       |    8.0    |   4.0   |    5.0     |    3.0    |
> | PoT      |    1.0    |   1.0   |    1.0     |    1.0    |
> | CoC      |    1.0    |   1.0   |    1.0     |    1.0    |
> | SC (N=3) |    3.0    |  24.0   |    15.0    |    9.0    |
> | SR (T=2) |    1.4    |   1.9   |    1.5     |    1.6    |
> | T=2, N=3 |    5.4    |   5.9   |    5.5     |    5.6    |
>
> |          | Deductive |               | Abductive |               |
> | -------- | :-------: | :-----------: | :-------: | :-----------: |
> |          | CRUXEval  | LiveCodeBench | CRUXEval  | LiveCodeBench |
> | Standard |    1.0    |      1.0      |    1.0    |      1.0      |
> | CoT      |    1.0    |      1.0      |    1.0    |      1.0      |
> | SC (N=3) |    3.0    |      3.0      |    3.0    |      3.0      |
> | SR (T=2) |    1.6    |      1.7      |    1.4    |      1.5      |
> | CoC      |    1.0    |      1.0      |    1.0    |      1.0      |
> | T=2, N=1 |    1.6    |      1.7      |    1.4    |      1.5      |
>
> **Total Cost** (cent)
>
> |          | List Func | MiniARC | RobustFill | Deepcoder |
> | -------- | :-------: | :-----: | :--------: | :-------: |
> | IO       |   10.2    |   4.6   |    2.0     |    3.3    |
> | PoT      |    5.0    |   3.7   |    0.6     |    1.2    |
> | CoC      | **11.0**  | **9.0** |    1.1     |    1.4    |
> | SC (N=3) |    5.3    |   3.7   |    0.6     |    1.2    |
> | SR (T=2) |    4.6    |   3.3   |    0.5     |    1.1    |
> | T=2, N=3 |    8.6    |   4.0   |  **3.1**   |  **4.7**  |
>
> |          | Deductive |               | Abductive |               |
> | -------- | :-------: | :-----------: | :-------: | :-----------: |
> |          | CRUXEval  | LiveCodeBench | CRUXEval  | LiveCodeBench |
> | Standard |    2.9    |      0.5      |    3.1    |      1.5      |
> | CoT      | **19.4**  |      3.7      | **19.5**  |      3.3      |
> | SC (N=3) |    2.9    |      0.5      |    3.3    |      1.5      |
> | SR (T=2) |    3.8    |      0.6      |    3.4    |      1.6      |
> | CoC      |   18.3    |      4.1      |   19.0    |      3.4      |
> | T=2, N=1 |   19.0    |    **4.4**    |   18.8    |    **4.4**    |
>
> From the results above, it can be seen that the RHDA pipeline is not a token-inefficient approach; it delivers significant performance gains within an acceptable cost range. Due to our limited understanding of specific advancements in non-symbolic domains, we cannot definitively conclude whether RHDA can generalize effectively to such domains. However, our work has conducted over 300 experiments with in-depth analysis and further extended the framework to the VirtualHome environment, demonstrating the robust generalizability of RHDA.
>
> [1] Madaan, Aman, et al. "Self-refine: Iterative refinement with self-feedback." *Advances in Neural Information Processing Systems* 36 (2024).
>
> [2] Huang, Jie, et al. "Large Language Models Cannot Self-Correct Reasoning Yet." *The Twelfth International Conference on Learning Representations*.

---

> ### Comment · Reviewer_JURK · 2024-11-22
>
> Thank the authors for their response! Most of the points make sense! I have raised the score!
>
> --For CoT, a 2-shot setting is employed for deductive and abductive reasoning tasks, while Sub-Hyp utilizes a zero-shot setting
>
> I think CoT can also be used in a zero-shot way?

---

> > ### Author Response · Authors · 2024-11-22
> >
> > Thank you for your acknowledgment and affirmation/confirmation! Of course, it is possible to use the zero-shot setting for CoT, but the reason we chose to use 2-shot is to ensure a fair comparison with other baselines, such as CoC. As for the RHDA method, due to its unique reflection process, we wanted to provide it with higher difficulty and more freedom in problem-solving to fully assess its capabilities. This design aims to highlight the potential advantages of the RHDA method under different conditions, while ensuring the comparability and credibility of the experimental results.
> >
> > If you have any other questions, please feel free to ask. Your valuable feedback has greatly enhanced the integrity of our work.

---

### Official Review · Reviewer_FKt4 · 2024-11-02

**Soundness:** 3
**Presentation:** 3
**Contribution:** 3
**Rating:** 6
**Confidence:** 4

**Summary:**

The paper presents "code reasoning" as a novel task requiring both reasoning and memory to explore the capabilities and boundaries of LLMs. The authors propose three new meta-benchmarks based on different forms of logical reasoning, namely: inductive code reasoning (predicting program from input-output pairs), deductive code reasoning (predicting output from input-program), and abductive code reasoning (predicting input from output-program). They instantiate these meta-benchmarks into eight concrete benchmarks: List Function, MiniARC, RobustFill, and DeepCoder for inductive reasoning, and CRUXEval and LiveCodeBench for both deductive and abductive reasoning. They propose a novel pipeline, "RHDA" (Reflective Hypothesis Decomposition and Amendment), that iteratively decomposes complex problems into simpler steps, verifies the results, and based on this feedback proposes amendments. This pipeline leads to as much as 3× performance gains over baseline models.

**Strengths:**

1. Novel formulation and pipeline:
    - The paper introduces an intermediate task between reasoning and recall, giving us the ability to understand LLMs more deeply
    - RHDA pipeline demonstrates robustness in handling complex reasoning tasks

2. Comprehensive Evaluation:
    - Three types of meta-framework with 8 specific benchmarks
    - Evaluation across different model types (GPT-4o and Claude 3.5)
    - Thorough ablation studies show the importance of each component.
3. Strong Results:
    - On inductive reasoning: RHDA outperforms baselines by 18.45% (List Function), 5.89% (MiniARC), 33.31% (RobustFill), and 12.02% (DeepCoder)
    - On deductive reasoning: Achieves 90.62% on CRUXEval and 84.16% on LiveCodeBench
    - On abductive reasoning: Achieves 83.75% on CRUXEval and 71.57% on LiveCodeBench
4. Extensibility: Successfully demonstrated application to VirtualHome environment, showing the framework can handle real-world complex scenarios like household tasks.

**Weaknesses:**

1. Limited Theoretical Analysis:
  - While the paper positions code reasoning between memory and reasoning tasks, it lacks a deeper theoretical analysis of why this intermediate position is beneficial or how it relates to existing theories of reasoning.
2. Limited Real-World Validation:
 - While they show the program is capable of performing in VirtualHome, having quantitative benchmarks and more complex scenarios would be helpful to understand real-world applicability.
3. Incomplete Error Analysis:
- Additional discussion on the failure modes of the RHDA pipeline would be helpful. Analysis of whether failures occur due to incorrect hypothesis formation, ineffective amendment etc is missing.
4. Complexity Overhead:
- The paper doesn't analyze the computational cost of running multiple iterations of RHDA compared to single-pass methods like CoT or PoT. This is particularly important given the iterative nature of the approach.

**Questions:**

1. Example analysis is said to be shown in Appendix E, but Appendix E is empty.
2. How sensitive is the method to the quality of the initial decomposition? Are there cases where poor initial decomposition cannot be recovered through amendments?
3. When and why does the RHDA pipeline fail - are the failures primarily due to poor initial decompositions, ineffective amendments, or do different tasks exhibit different failure patterns?
4. The computational overhead analysis is missing:
 -  What is the computational cost compared to baseline methods?
 -  What is the trade-off between the number of iterations and performance gains?
5. While VirtualHome examples demonstrate RHDA's potential for real-world tasks, can you provide quantitative metrics and more complex scenarios (e.g., handling multiple interdependent tasks or recovering from failures) to better evaluate its practical capabilities?

---

> ### Author Response · Authors · 2024-11-22
> **Official Response to Reviewer FKt4 Part (1/3)**
>
> We sincerely thank Reviewer FKt4 for their constructive suggestions. Below is our detailed response:
>
> > While the paper positions code reasoning between memory and reasoning tasks, it lacks a deeper theoretical analysis of why this intermediate position is beneficial.
>
> We sincerely thank the reviewer for thoughtful comments and for highlighting the need for a more in-depth theoretical analysis. Our conclusion stems from an interesting observation: incorporating code data into the pretraining phase can improve performance on reasoning tasks beyond just coding tasks [1, 2, 3]. The reasons behind this phenomenon remain unclear, and the LLM community currently lacks sufficient  empirical evidence to explain why this occurs, making it challenging to prove theoretically.
>
>  We hypothesize that code, while strictly adhering to syntax rules, also embodies inherent reasoning potential. Enhancing code-related capabilities during the pretraining stage could potentially address complex reasoning tasks and further facilitate "memory-based reasoning." This work seeks to provide empirical evidence evidence  supporting this hypothesis, though a formal theoretical framework is beyond its scope. We hope that our experiments will inspire further research in this direction.
>
> > Having quantitative benchmarks and more complex scenarios would be helpful to understand real-world applicability. Can you provide quantitative metrics about VirtualHome .
>
> To the best of our knowledge, adequate automated evaluation methods for the VirtualHome environment remain unavailable, which is a main reason the original paper does not provide quantitative metrics for this scenario. To validate the effectivenWess of the RHDA method, we **manually** evaluated the execution failure rate across 52 tasks in two scenarios. The experimental results, presented in the accompanying table, demonstrate that the native GPT-4o encounters significant challenges in adapting to VirtualHome. The primary cause of failure lies in generating scripts that, while semantically similar to correct actions, are not executable within the environment (e.g., `open the tap` is invalid action, whereas `touch the tap` is valid action). By employing the RHDA method, which incorporates step-by-step solutions and effective feedback mechanisms, the error rate was significantly reduced. We have added the relevant discussion to the latest version of the paper in Appendix D.
>
> |                                  | native GPT-4o | w/o Sub-Hyp | w/o Amend |   RHDA   |
> | -------------------------------- | :-----------: | :---------: | :-------: | :------: |
> | # Error Action $\downarrow$      |      92       |     84      |    84     |  **52**  |
> | Avg. Error per Step $\downarrow$ |     0.84      |    0.35     |   0.20    | **0.16** |
> | Avg. Err per Task $\downarrow$   |     2.09      |    1.83     |   1.75    | **1.08** |
>
>
>
> > Incomplete Error Analysis: Additional discussion on the failure modes of the RHDA pipeline would be helpful. Analysis of whether failures occur due to incorrect hypothesis formation, ineffective amendment etc is missing.
>
> Thank you for your suggestion. We have added the relevant discussion in Section 4.4 and Appendix E.2 in paper. More detailed discussions are included in our responses to subsequent questions.
>
> > Complexity Overhead: The paper doesn't analyze the computational cost of running multiple iterations of RHDA compared to single-pass methods like CoT or PoT. This is particularly important given the iterative nature of the approach.
>
> We agree with this point and have added the relevant discussion in Appendix F and our responses to subsequent questions.
>
> > Appendix E is empty.
>
> This is a minor formatting issue: Appendix E originally lacked textual content, and its associated table appeared on the subsequent page. Consequently, the titles of Appendices E and F were displayed together, creating the impression that Appendix E was empty. To address this, we added appropriate textual content and optimized the layout to ensure that readers can easily locate the content of Appendix E.  These revisions have been included in the latest version of the paper.

---

> ### Author Response · Authors · 2024-11-22
> **Official Response to Reviewer FKt4 Part (2/3)**
>
> > How sensitive is the method to the quality of the initial decomposition? Are there cases where poor initial decomposition cannot be recovered through amendments?
>
> Insightful question. In our experiments, we indeed observed that LLMs are sensitive to their initial hypotheses, often failing to explore the correct solution path even after multiple amendments. We believe this issue arises due to two primary reasons:
>
> 1. **Intrinsic limitations of LLMs or task complexity**: This suggests that LLMs struggle to solve problems that exceed their inherent capabilities.
> 2. **Misleading initial hypotheses**: When the initial hypothesis is flawed, subsequent amendments are unable to break the LLM out of its fixed thinking patterns to explore new hypotheses.
>
> The underlying cause may lie in the sparsity of data in LLM's pre-training stage, leading to insufficient understanding of code reasoning tasks. We provide an example demonstrating how amendments failed to recover from an initial flawed hypothesis, with additional examples detailed in Appendix E.2.
>
> ```List function Failure Case
> Observations:
> [3, 4, 1, 5, 2, 0, 8, 6, 9]->[1]
> [5, 0, 6, 8, 2, 9, 4, 7, 3]->[6]
> [6, 3, 1, 4, 9, 0, 7]->[1]
> [9, 5, 7, 2]->[7]
> Initial Hypothesis:
> Step1: Identify the smallest integer in the input list.
> Step2: Find the index of the first occurrence of this smallest integer.
> Step3: Check if there is an integer immediately following this smallest integer in the list.
> Step4: If such an integer exists and is larger than the smallest integer, return it.
> Step5: If the smallest integer is the last element in the list or the next integer is not larger, return the smallest integer itself.
>
> Execution and Amendment Submission... (ignored)
>
> Recovered Hypothesis:
> Step1: Identify the smallest integer in the input list.
> Step2: Find the index of the first occurrence of this smallest integer.
> Step3: From this index onward, find the largest integer in the list.
> Step4: If the smallest integer is the last element in the list, return the smallest integer itself.
> Step5: Otherwise, return the largest integer found after the smallest integer.
> ```
>
> In this example, we observed that the LLM became fixated on the hint of finding the smallest integer, repeatedly attempting variations of this approach to solve the problem. However, if it had been able to break out of this fixed thinking pattern and instead counted the position of the output number in the list, the problem could have been resolved.
>
> > When and why does the RHDA pipeline fail - are the failures primarily due to poor initial decompositions, ineffective amendments, or do different tasks exhibit different failure patterns?
>
> The RHDA pipeline leverages the inherent capabilities of LLMs to explore problem-solving pathways; however, its effectiveness is significantly constrained for tasks that exceed the LLM's capacity. As the reviewer correctly observed, RHDA failures often stem from the breakdown of initial hypotheses. This issue can be attributed to the intrinsic limitations of LLMs: the failure of initial hypotheses suggests that the LLM lacks the fundamental ability to solve the problem, while the amendment submission merely provides superficial adjustments to observed errors. Finally, the analysis and reflection on such failures remain dependent on the LLM's reasoning capabilities. The limited impact of corrective feedback further underscores the inadequacy of LLMs in addressing these complex challenges.
>
> > What is the computational cost compared to baseline methods?
>
> We calculated the average number of API calls and the total cost associated with using GPT-4o. While our approach incurs slightly higher costs compared to PoT, it proves to be more cost-effective for specific tasks, highlighting its efficiency in targeted applications.
>
> **Avg. API Calls**
>
> |          | List Func | MiniARC | RobustFill | Deepcoder |
> | -------- | :-------: | :-----: | :--------: | :-------: |
> | IO       |    8.0    |   4.0   |    5.0     |    3.0    |
> | PoT      |    1.0    |   1.0   |    1.0     |    1.0    |
> | CoC      |    1.0    |   1.0   |    1.0     |    1.0    |
> | SC (N=3) |    3.0    |  24.0   |    15.0    |    9.0    |
> | SR (T=2) |    1.4    |   1.9   |    1.5     |    1.6    |
> | T=2, N=3 |    5.4    |   5.9   |    5.5     |    5.6    |
>
> |          | Deductive |               | Abductive |               |
> | -------- | :-------: | :-----------: | :-------: | :-----------: |
> |          | CRUXEval  | LiveCodeBench | CRUXEval  | LiveCodeBench |
> | Standard |    1.0    |      1.0      |    1.0    |      1.0      |
> | CoT      |    1.0    |      1.0      |    1.0    |      1.0      |
> | SC (N=3) |    3.0    |      3.0      |    3.0    |      3.0      |
> | SR (T=2) |    1.6    |      1.7      |    1.4    |      1.5      |
> | CoC      |    1.0    |      1.0      |    1.0    |      1.0      |
> | T=2, N=1 |    1.6    |      1.7      |    1.4    |      1.5      |

---

> ### Author Response · Authors · 2024-11-22
> **Official Response to Reviewer FKt4 Part (3/3)**
>
> **Total Cost** (cent)
>
> |          | List Func | MiniARC | RobustFill | Deepcoder |
> | -------- | :-------: | :-----: | :--------: | :-------: |
> | IO       |   10.2    |   4.6   |    2.0     |    3.3    |
> | PoT      |    5.0    |   3.7   |    0.6     |    1.2    |
> | CoC      | **11.0**  | **9.0** |    1.1     |    1.4    |
> | SC (N=3) |    5.3    |   3.7   |    0.6     |    1.2    |
> | SR (T=2) |    4.6    |   3.3   |    0.5     |    1.1    |
> | T=2, N=3 |    8.6    |   4.0   |  **3.1**   |  **4.7**  |
>
> |          | Deductive |               | Abductive |               |
> | -------- | :-------: | :-----------: | :-------: | :-----------: |
> |          | CRUXEval  | LiveCodeBench | CRUXEval  | LiveCodeBench |
> | Standard |    2.9    |      0.5      |    3.1    |      1.5      |
> | CoT      | **19.4**  |      3.7      | **19.5**  |      3.3      |
> | SC (N=3) |    2.9    |      0.5      |    3.3    |      1.5      |
> | SR (T=2) |    3.8    |      0.6      |    3.4    |      1.6      |
> | CoC      |   18.3    |      4.1      |   19.0    |      3.4      |
> | T=2, N=1 |   19.0    |    **4.4**    |   18.8    |    **4.4**    |
>
> All the results mentioned above have been updated in Appendix F of the latest version of our paper.
>
> > What is the trade-off between the number of iterations and performance gains?
>
> In our experiments, we observed a consistent improvement in the performance of all datasets for inductive reasoning tasks, indicating that the LM can iteratively refine its sub-hypotheses. This suggests that using an LM with a longer context length allows for further improvement through additional iterations. In contrast, for deductive reasoning tasks, where the performance is already quite good, two iterations suffice to reach optimal results, and more iterations may not bring additional benefits, potentially even leading to slight degradation in performance. For abductive reasoning tasks, we found that as the number of iterations increased, the performance improved, but the rate of improvement gradually slowed down, exhibiting a clear marginal effect.
>
> |      | Accuracy  |         |            |           |      | Task Accuracy |         |            |           |
> | ---- | :-------: | :-----: | :--------: | :-------: | ---- | ------------- | ------- | ---------- | --------- |
> |      | List Func | MiniARC | RobustFill | Deepcoder |      | List Func     | MiniARC | RobustFill | Deepcoder |
> | T=1  |   47.10   |  8.46   |   35.65    |   30.21   | T=1  | 47.10         | 8.46    | 35.65      | 19.79     |
> | T=2  |   51.05   |  12.56  |   43.48    |   38.89   | T=2  | 51.05         | 12.56   | 43.48      | 23.96     |
> | T=3  |   53.20   |  14.10  |   47.83    |   38.19   | T=3  | 53.20         | 14.10   | 47.83      | 26.04     |
>
> |      | Deductive |               | Abductive |               |
> | ---- | :-------: | :-----------: | :-------: | :-----------: |
> |      | CRUXEval  | LiveCodeBench | CRUXEval  | LiveCodeBench |
> | T=1  |   86.62   |     71.29     |   77.12   |     60.78     |
> | T=2  |   90.62   |     84.16     |   83.75   |     71.57     |
> | T=3  |   90.62   |     82.35     |   87.00   |     75.49     |
>
> We have added the relevant discussion in Appendix G of the latest version of our paper.
>
> [1] Aryabumi, Viraat, et al. "To Code, or Not To Code? Exploring Impact of Code in Pre-training." *arXiv preprint arXiv:2408.10914* (2024).
>
> [2] Yang, Ke, et al. "If LLM Is the Wizard, Then Code Is the Wand: A Survey on How Code Empowers Large Language Models to Serve as Intelligent Agents." *ICLR 2024 Workshop on Large Language Model (LLM) Agents*.
>
> [3] MA, YINGWEI, et al. "At Which Training Stage Does Code Data Help LLMs Reasoning?." *ICLR 2024*

---

> > ### Comment · Reviewer_FKt4 · 2024-11-27
> >
> > I thank the authors for their detailed responses to all my concerns!

---

### Official Review · Reviewer_773h · 2024-11-03

**Soundness:** 2
**Presentation:** 2
**Contribution:** 2
**Rating:** 5
**Confidence:** 3

**Summary:**

This article presents a dynamic prompting method that automatically adjusts the number of prompting steps based on task complexity and model performance in real time.

**Strengths:**

The approach taken has significant potential. The provided benchmark can be instrumental in understanding the boundaries between reasoning and recall.

**Weaknesses:**

The paper contains  arguments that "are a bit loose or controversial". I request that the authors make their arguments more precise, and address the ICLR audience rather than assuming that this audience knows cognitive science, and terms like System 1 and System 2 tasks.

It is helpful to learn that System 1 and System 2 tasks have been explored technically, but the authors must introduce these tasks from the ML perspective and provide the references  [5, 6, 7] to ensure readers have the proper context.

  From a computer science point of view, an LLM can be viewed either as performing a mathematical function or as a software artifact.  Claims made about LLMs thus need to be validated either mathematically or empirically. The responses in the rebuttal help significantly to make things more precise.

Further, the goal to "explore the boundaries of LLM capabilities" can be presented in a more precise way. Many ML practitioners don't care about whether an LLM corresponds to some theoy of mind or human reasoning, but want to know in a precise sense what it's capabilities are.

The paper starts off in line 42 with "From the perspective of human cognitive psychology, reasoning can be viewed as a process of memory retrieval," and this is the perspective taken. There is insufficient mathematical theory or implementation to show results relevant to ICLR.

There are several claims made without defintion and/or validation. Some of these issues have been addressed in the rebuttal, but the authors must revise the article paying attention to ensuring clarity for an ML researcher.

**Questions:**

Can you provide any empirical evidence to back up the many claims made in the article?

---

> ### Author Response · Authors · 2024-11-15
> **Official Response to Reviewer 773h Part (1/2)**
>
> We would like to thank Reviewer 773h for the review.
>
> It seems that the reviewer has misunderstood our work. We understand this and sincerely appreciate the time the Reviewer 773h has invested. The summary and strengths highlighted by the reviewer do not align with the core contributions of our work, and we respectfully disagree with some of the weakness pointed out. We will address each point of misunderstanding from Reviewer 773h and hope that the reviewer will reassess the quality of our paper after we have made our best efforts to clarify our work.
>
> We would like to clarify that one of the expected contributions of our work is to draw the LLM community’s attention to the task of “code reasoning.” A substantial amount of research has demonstrated [1, 2, 3, 4] that incorporating code data during the pre-training and fine-tuning stages can improve performance on reasoning tasks beyond just coding tasks. This counterintuitive phenomenon has prompted our thinking, and we believe that utilizing code to enhance LLMs’ reasoning capabilities is a promising approach. LLMs have a strong memorization ability but are not particularly adept at reasoning, so leveraging their ability to “recall” code and “reason” with it could be beneficial.
>
> Given that ICLR is an open, inclusive, and responsible top machine learning conference, we have attempted to offer a unified explanation of LLMs’ “recall” and “reasoning” from the perspective of cognitive psychology. This does not imply that our research is unrelated to ICLR. We have proposed an intriguing code reasoning benchmark and a novel method, RHDA, which has achieved high performance and garnered support from Reviewer FKt4 and Reviewer JURK. This is highly aligned with the ICLR theme of "foundation and frontier models, including LLMs".
>
> > LLMs perform better on System 1 tasks than on System 2 tasks. ???? Must define System 1 tasks and System 2 tasks for this to make any sense.
>
> System 1 and System 2 are concepts derived from cognitive theory and have been widely introduced in the field of deep learning [5, 6, 7]. As depicted in Figure 1 of our original paper, System 1 is a fast, automatic, and intuitive mode used for solving simple tasks, while System 2 is a slow, logical mode used for addressing complex reasoning problems.
>
> >We believe that, similar to humans, there is no significant distinction between memorizing and reasoning tasks for LLM
> >
> >----this is a scientific document NOT a statement of unprovable beliefs.
>
> In the original text, the phrase “We believe” is intended to articulate our observations and hypotheses regarding the current capabilities of LLM in handling memory and reasoning tasks, rather than an unprovable personal belief. For example, [8, 9, 10] explore the similarities between humans and LLMs, with [8] specifically emphasizing that “LLMs exhibit behaviour that is consistent with the outputs of mentalistic inference in humans.” These studies provide support for our perspective. Similarly, [11, 12] provide empirical evidence showing that LLMs require the collaboration of reasoning and memorization to perform mathematical reasoning. At last, we would like to clarify that we have already provided substantial empirical evidence for our core contributions, while the assertions cited by Reviewer 773h are not part of our core contribution and therefore do not require empirical evidence in original paper.

---

> > ### Comment · Reviewer_773h · 2024-11-19
> > **Rebuttal has changed my views.**
> >
> > Thanks to the authors for their detailed rebuttal. This response has changed my opinions of the article and I am happy to revise my review appropriately. I better understand the goal: to create a unified explanation of LLMs’ “recall” and “reasoning” from the perspective of cognitive psychology.
> >
> > It is helpful to learn that System 1 and System 2 tasks have been explored technically, but the authors must introduce these tasks from the ML perspective and provide the references  [5, 6, 7] to ensure readers have the proper context.
> >
> > I request that the authors make their arguments more precise. All reviewers agree thath the paper contains  arguments that "are a bit loose or controversial". From a computer science point of view, an LLM can be viewed either as performing a mathematical function or as a software artifact.  Claims made about LLMs thus need to be validated either mathematically or empirically. The responses in the rebuttal help significantly to make things more precise.
> >
> > Further, the goal to "explore the boundaries of LLM capabilities" can be presented in a more precise way. Many ML practitioners don't care about whether an LLM corresponds to some theoy of mind or human reasoning, but want to know in a precise sense what it's capabilities are.

---

> > > ### Author Response · Authors · 2024-11-25
> > > **Further Clarification and Enhancement.**
> > >
> > > We sincerely appreciate your understanding and support! It is truly encouraging for us to clearly articulate the motivation behind our work and to gain your attention and recognition.
> > >
> > > Below is a summary of the thought process behind our introduction section: we begin by discussing the unification of memory and reasoning from the perspective of cognitive psychology and draw an analogy to LLMs. Additionally, we propose a novel task—code reasoning—which possesses characteristics of both memory and reasoning. Exploring this task can help practitioners in the field of ML better identify the limitations of existing LLMs and further investigate potential pathways toward achieving "memory-based reasoning."
> > >
> > > In response to your feedback, we have updated the introduction section accordingly and included additional relevant references to provide a clearer understanding of the research background for ML practitioners. These references encompass numerous empirical studies that demonstrate how LLMs' reasoning capabilities are largely derived from memory. However, a theoretical discussion of this phenomenon lies beyond the scope of this paper and has thus not been elaborated upon.
> > >
> > > If you have any further questions or suggestions, please do not hesitate to contact us. Your evaluation and valuable feedback are of great importance to the improvement of our work!

---

> ### Author Response · Authors · 2024-11-15
> **Official Response to Reviewer 773h Part (2/2)**
>
> > what is "the execution space"?
>
> The execution space refers to the set of all possible operations that a program can perform. In this space, each point represents a possible program that implements a particular function.
>
> Specifically, a neural program synthesis model searches through this function execution space in an attempt to find a program that can satisfy the input-output example requirements.
>
> > what is "abstract level understanding of function’s behavior" or "perform abductive inference" ?
>
> The "abstract level understanding of function's behavior" refers to the ability of an LLM to comprehend the function's functionality given its output and program, even in situations where execution is not possible. This is assessing whether the LLM is capable of reasonably inferring the cause (input) after observing the result (output).
>
> > Can you provide any empirical evidence to back up the many claims made in the article?
>
> The core contribution of our paper is the definition of the code reasoning task to explore the boundaries of LLM capabilities. We define three meta-benchmarks and concretize them into eight specific benchmarks. Additionally, we propose a novel method, the Reflective Hypothesis Decomposition and Amendment (RHDA) pipeline. We have conducted over 100 experiments on the state-of-the-art models GPT-4o and Claude3.5 to validate the effectiveness of our method. The concerns raised by Reviewer 773h do not pertain to our core contribution and do not require empirical evidence.
>
> [1] Aryabumi, Viraat, et al. "To Code, or Not To Code? Exploring Impact of Code in Pre-training." *arXiv preprint arXiv:2408.10914* (2024).
>
> [2] Yang, Ke, et al. "If LLM Is the Wizard, Then Code Is the Wand: A Survey on How Code Empowers Large Language Models to Serve as Intelligent Agents." *ICLR 2024 Workshop on Large Language Model (LLM) Agents*.
>
> [3] MA, YINGWEI, et al. "At Which Training Stage Does Code Data Help LLMs Reasoning?." *ICLR 2024*
>
> [4] Zhang, Xinlu, et al. "Unveiling the Impact of Coding Data Instruction Fine-Tuning on Large Language Models Reasoning." *arXiv preprint arXiv:2405.20535* (2024).
>
> [5] Yoshua Bengio, “From System 1 Deep Learning to System 2 Deep Learning.” https://nips.cc/Conferences/2019/Schedule?showEvent=15488
>
> [6] Yao, Shunyu, et al. "Tree of thoughts: Deliberate problem solving with large language models." *Advances in Neural Information Processing Systems* 36 (2024).
>
> [7] Weston, Jason, and Sainbayar Sukhbaatar. "Syste m 2 Attention (is something you might need too)." *arXiv preprint arXiv:2311.11829* (2023).
>
> [8] Strachan, James WA, et al. "Testing theory of mind in large language models and humans." *Nature Human Behaviour* (2024): 1-11.
>
> [9] Shanahan, Murray, Kyle McDonell, and Laria Reynolds. "Role play with large language models." *Nature* 623.7987 (2023): 493-498.
>
> [10] Hutson, Matthew. “How does ChatGPT 'think'? Psychology and neuroscience crack open AI large language models.” *Nature* vol. 629,8014 (2024): 986-988. doi:10.1038/d41586-024-01314-y
>
> [11] Schaeffer, Rylan, Brando Miranda, and Sanmi Koyejo. "Are emergent abilities of large language models a mirage?." Advances in Neural Information Processing Systems 36 (2024).
>
> [12] Razeghi, Yasaman, et al. "Impact of Pretraining Term Frequencies on Few-Shot Numerical Reasoning." Findings of the Association for Computational Linguistics: EMNLP 2022. 2022.

---

### Author Response · Authors · 2024-11-25
**General Response**

We sincerely thank each reviewer for their thorough and valuable feedback. Guided by the reviewers' comments, we have incorporated additional experiments and discussions. While ensuring full consistency with the original contributions and conclusions of the paper, we have made appropriate modifications to enhance its comprehensiveness. These revisions have been highlighted in blue for clarity. Below is a brief summary of the main changes we made:

| Changes | Section | Related Reviewers |
|---|---|---|
| More Base LLM: Llama3.1-70B-Instruct and Qwen-max | Appendix B | Reviewer JURK |
| Additional Baseline:Self Refine | Section 4 | Reviewer JURK |
| Human evaluation results on VirtualHome | Appendix D | Reviewer FKt4,Reviewer JURK |
| Layout optimization and more case studies | Appendix E.1 | Reviewer FKt4 |
| Failure analyse | Section 4 and Appendix E.2 | Reviewer FKt4 |
| Additional Cost Report | Appendix F | Reviewer FKt4,Reviewer JURK |
| Additional analysis of accuracy over iterations | Appendix G | Reviewer FKt4 |
| More references for recall and reasoning | Section 1 | Reviewer 773h, Reviewer FKt4,Reviewer JURK |

All additional experiments are included in the appendix, which is organized as follows:

- Appendix A: Detailed description of the DSL syntax.
- Appendix B: Extended experimental results for more LLM base models.
- Appendix C: Specific details of the benchmarks.
- Appendix D: Qualitative and quantitative analyses of the RHDA process framework on VirtualHome.
- Appendix E: Presentation and analysis of both successful and failure cases.
- Appendix F: Detailed explanation of experimental costs.
- Appendix G: Trade-off analysis between the number of iterations and performance gains.
- Appendix H: Prompts used in the experiments.

If you have additional questions, please let us know.

---

### Meta-Review · Area_Chair_L41B · 2024-12-20

**Metareview:**

This paper concerns code reasoning using LLMs. Thereby, the LLMs require both recall and reasoning capabilities, rendering the underlying task rather complex.

In more detail, the paper presents "code reasoning" as a novel task requiring both reasoning and recall to explore the capabilities and boundaries of LLMs. Three tasks of logical reasoning are presented: inductive code reasoning (predicting program from input-output pairs), deductive code reasoning (predicting output from input-program), and abductive code reasoning (predicting input from output-program).

These tasks yield eight concrete benchmarks: List Function, MiniARC, RobustFill, and DeepCoder for inductive reasoning, and CRUXEval and LiveCodeBench for both deductive and abductive reasoning. The novel pipeline "RHDA" (Reflective Hypothesis Decomposition and Amendment) iteratively decomposes such complex problems into simpler steps, verifies the results, and proposes amendments. This pipeline leads to as much as 3× performance gains over baseline models.

The strengths of the paper as identified by the reviewers center around the novel problem formulation to understand LLMs' complex reasoning in a better way, with a good benchmark evaluation across different LLM models. Even real-world scenarios are not out of reach. The paper is generally very well written.

The reviewers have, in addition, listed several weaknesses, such as more precise explanations and definitions or a missing real-world evaluation. Yet, after the discussion phase, the reviewers agreed that the paper might be accepted.

**Additional Comments On Reviewer Discussion:**

The discussion phase was intense, with good engagement of both reviewer and authors, leading to an improved understanding of the paper.

---

### Decision · Program_Chairs · 2025-01-22

Accept (Poster)